# Phytochemistry and Biological Studies of Endemic Hawaiian Plants

**DOI:** 10.3390/ijms242216323

**Published:** 2023-11-14

**Authors:** Pornphimon Meesakul, Tyler Shea, Roland Fenstemacher, Shi Xuan Wong, Yutaka Kuroki, Aya Wada, Shugeng Cao

**Affiliations:** 1Department of Pharmaceutical Sciences, Daniel K. Inouye College of Pharmacy, University of Hawai’i at Hilo, 200 W. Kawili St., Hilo, HI 96720, USA; pmeesak@hawaii.edu; 2Chemistry Department, University of Hawai’i at Hilo, 200 W. Kawili St., Hilo, HI 96720, USA; tylerms3@hawaii.edu; 3Chemistry Laboratory, Board of Water Supply, City and County of Honolulu, 630 South Beretania Street, Honolulu, HI 96843, USA; hale_noa@yahoo.com; 4Delightex Pte. Ltd., 230 Victoria Street, #15-01/08 Bugis Junction Towers, Singapore 188024, Singapore; shixuan@delightexplorers.com (S.X.W.); yutaka@delightexplorers.com (Y.K.); aya@delightexplorers.com (A.W.)

**Keywords:** Hawaiian plants, endemic Hawaiian plant, phytochemistry, chemical constituents, biological studies, biological activities

## Abstract

The Hawaiian Islands are renowned for their exceptional biodiversity and are host to a plethora of endemic plant species, which have been utilized in traditional Hawaiian medicine. This scientific review provides an in-depth analysis of the phytochemistry and biological studies of selected endemic Hawaiian plants, highlighting their medicinal properties and therapeutic potential. A literature search was conducted, utilizing major academic databases such as SciFinder, Scopus, Web of Science, PubMed, Google Scholar, Science Direct, and the Scientific Information Database. The primary objective of this search was to identify relevant scholarly articles pertaining to the topic of the review, which focused on the phytochemistry and biological studies of endemic Hawaiian plants. Utilizing these databases, a comprehensive range of literature was obtained, facilitating a comprehensive examination of the subject matter. This review emphasizes the rich phytochemical diversity and biological activities found in Endemic Hawaiian plants, showcasing their potential as sources of novel therapeutic agents. Given the unique biodiversity of Hawaii and the cultural significance of these plants, continued scientific exploration, conservation, and sustainable utilization of these valuable resources is necessary to unlock the full potential of these plant species in drug discovery and natural product-based therapeutics.

## 1. Introduction

Hawaii, an archipelago situated in the central Pacific Ocean, is renowned for its unique biodiversity and distinctive ecosystems [1,2,3,4,5]. The isolated nature of the islands, combined with the unique geological and climatic conditions, has given rise to a rich array of endemic plant species found nowhere else in the world [1,2,3,4,5]. These remarkable plants have not only fascinated scientists and naturalists but have also captivated the interest of researchers in the fields of phytochemistry and biology. Endemic Hawaiian plants, with their diverse chemical profiles and exceptional biological properties, have become the focus of numerous investigations aimed at unravelling their phytochemical composition and exploring their potential applications in various fields, including medicine, agriculture, and industry. The phytochemical studies of these remarkable botanical treasures have provided valuable insights into the chemical constituents responsible for their distinct characteristics and biological activities. This review article aims to provide a comprehensive overview of the current state of knowledge regarding the phytochemical and biological studies of endemic Hawaiian plants, referring to “A Review of the Endemic Genera of Hawaiian Plants” reported by Benjamin in 1967 [5], “Flora of the Hawaiian Islands” by Warren L. Wagner, the Department of Botany, Smithsonian Institution [6], and the “Hawaiian Ethnobotany Online Database” by Bishop Museum [7]. By analyzing the available literature, we will delve into the diverse natural products isolated from these unique plant species, highlighting their structural features and biological activities. Moreover, studies have elucidated the mechanisms underlying these bioactive properties, providing a foundation for future drug discovery and development efforts. By collecting and critically analyzing the existing literature, this review aims to consolidate the vast body of knowledge on the phytochemistry and biological studies of Endemic Hawaiian plants. In doing so, it seeks to highlight the potential applications of these plants in various domains, while identifying gaps in our current understanding. Furthermore, this review will discuss the challenges and opportunities associated with the conservation and sustainable utilization of these unique botanical resources, emphasizing the importance of preserving Hawaii’s rich plant heritage for future generations.

## 2. Discussions

The Hawaiian Islands are known for their unique and diverse flora, characterized by an extraordinary number of endemic plant species found nowhere else in the world [1,2,3,4,5]. These endemic species have captured the interest of botanists and researchers for decades, and their study has shed light on the evolutionary processes and ecological dynamics that have shaped the Hawaiian archipelago [1,2,3,4,5].

The endemic plant checklist for Hawaii is documented in multiple scholarly sources, including Benjamin C. Stone’s 1976 work entitled “A Review of the Endemic Genera of Hawaiian Plants”, [5] the digital platform resource “Flora of the Hawaiian Islands” provided by Warren L. Wagner, the Department of Botany, Smithsonian Institution [6], and the “Hawaiian Ethnobotany Online Database” by Bishop Museum [7]. “A Review of the Endemic Genera of Hawaiian Plants” published in 1976 by Benjamin C. Stone is a valuable resource for understanding the unique genera of plants that are endemic to Hawaii. It provides detailed information about these endemic genera, including their characteristics, distribution, and significance in Hawaiian flora. Stone provided the information about the number of plant genera that are endemic to the Hawaiian Islands [5]. The “Flora of the Hawaiian Islands” website by the Department of Botany, Smithsonian Institution, is a comprehensive and up-to-date digital database of Hawaiian plant species. It should contain detailed information on the various plant species found in Hawaii, including 1232 plants [6]. The “Hawaiian Ethnobotany Online Database”, Bishop Museum, as previously mentioned, is a database which focuses on the ethnobotanical aspects of Hawaiian plants, emphasizing their traditional uses by native Hawaiians, thus offering valuable insights into the cultural and historical significance of these plants in Hawaiian society. These sources collectively contribute to our understanding of Hawaiian flora, including 467 plant species. Researchers and enthusiasts can use them to explore the botanical diversity and cultural significance of plants in the Hawaiian Islands [7].

In this review, we delve into the comprehensive work of three endemic Hawaiian plant checklists (Appendix A) [5] in terms of phytochemistry and biological activities (Appendix A). Stone’s article provides a fascinating exploration of the endemic genera, highlighting their significance and shedding light on the critical need for their conservation. However, a limited number of scientific publications exist pertaining to the phytochemistry and biological activities of endemic Hawaiian plants. The following summaries provide an overview of the research conducted in these areas. Our literature search yielded information on 73 endemic Hawaiian plant species that have been the subject of research and documentation pertaining to their phytochemical composition and biological activities, the details of which are summarized in Appendix A.

### 2.1. Argemone glauca *L. ex Pope, var.* glauca

*Argemone glauca* var. *glauca*, known as prickly poppy, belongs to the Papaveraceae or poppy family. The stems have stout, reflexed prickles, and spines, with around 5–25 per cm^2^ of surface. The lower leaf surfaces have about three to six prickles per cm^2^ on the larger veins, while each sepal boasts 12–30 prickles. On the capsules, there are roughly 35–65 spines per valve [8]. *A. glauca* var. *glauca* discovered on Lanai Island, Hawaii, was determined to possess a significant number of alkaloids, including protopine, allocryptopine, sanguinarine, berberine, and chelerythrine (Figure 1). These alkaloids align with what is typically found in a more specialized species of *Argemone* [9].

### 2.2. Bidens *Genus*

*Bidens*, a member of the Asteraceae family, is widely recognized as one of the most prominent examples of adaptive radiation within the Hawaiian flora. Remarkably, the 19 endemic *Bidens* species found in the Hawaiian archipelago exhibit a significantly greater range of ecological and morphological diversity compared with the approximately 230 species within the genus that are distributed across five continents. These endemic Hawaiian *Bidens* species are distributed across all eight main islands and span an extensive elevational gradient, thriving from sea level to altitudes exceeding 2200 m. These species occupy a diverse array of habitats including, but not limited to, sandy dunes, lava flows, arid deserts, scrublands, mesic forests, lush rainforests, and wetland bogs. It is worth noting that the majority of these taxa are exclusive to individual islands, with many further restricted to specific habitat types within or between islands. Additionally, a significant number of species and subspecies within the *Bidens* genus in Hawaii face urgent conservation concerns, being categorized as threatened, endangered, or critically endangered [10,11].

In 1984, Marchant and colleagues investigated polyacetylenes in both leaves and roots across 19 distinct species and six subspecies of Hawaiian *Bidens* (*B. amplectens*, *B. asymmetrica*, *B. campylotheca* subsp. *campylotheca*, *B. campylotheca* subsp. *pentamera*, *B. cervicata*, *B. conjuncta*, *B. cosmoides*, *B. forbesii* subsp. *forbesii*, *B. forbesii* subsp. *kahiliensis*, *B. hawaiensis*, *B. macrocarpa*, *B. mauiensis*, *B. menziesii* subsp. *filiformis*, *B. menziesii* subsp. *menziesii*, *B. micrantha* subsp. *ctenophylla*, *B. micrantha* subsp. *kalealaha*, *B. micrantha* subsp. *micrantha*, *B. molokaiensis*, *B. populifolia*, *B. sandvicensis* subsp. *confusa*, *B. sandvicensis* subsp. *sandvicensis*, *B. torta*, *B. valida*, and *B. wiebkei*). Among the compounds identified were 11 C_13_ hydrocarbons, aromatic and thiophenyl derivatives, one C_14_ tetrahydropyrene, and three C_17_ hydrocarbons (Figure 2), all of which can be traced back to a common precursor, oleic acid. Notably, polyacetylenes were absent in the leaves of 13 of these taxa, despite their presence in the roots of all those tested. Additionally, the discovery of 2-[2-phenyl-ethyne-1-yl]-5-acatoxymethyl thiophene (Figure 2) in *Bidens* represents a novel finding, as it has not been previously reported [11]. Its widespread occurrence aligns with other evidence suggesting that all Hawaiian *Bidens* species share a common ancestry from a single introduction to the islands. Furthermore, the complement of polyacetylenes in the roots and leaves proved to be a distinguishing feature for most taxa, with the exception of *B. torta*, where each population displayed a unique array of polyacetylenes. At a taxonomic level beyond species, no discernible pattern emerged in the distribution of polyacetylenes within this group [12].

In 1990, Ganders et al. studied 19 species and eight subspecies of *Bidens* (*B. amplectens*, *B. asymmetrica*, *B. campylotheca* subsp. *campylotheca*, *B. campylotheca* subsp. *pentamera*, *B. campylotheca* subsp. *waihoiensis*, *B. cervicata*, *B. conjuncta*, *B. cosmoides*, *B. forbesii* subsp. *forbesii*, *B. forbesii* subsp. *kahiliensis*, *B. hawaiensis*, *B. hillebrandiana*, *B. hillebrandiana* subsp. *polycephala*, *B. macrocarpa*, *B. mauiensis*, *B. menziesii* subsp. *filiformis*, *B. menziesii* subsp. *menziesii*, *B. micrantha* subsp. *ctenophylla*, *B. micrantha* subsp. *kalealaha*, *B. micrantha* subsp. *micrantha*, *B. molokaiensis*, *B. populifolia*, *B. sandvicensis* subsp. *confusa*, *B. sandvicensis* subsp. *sandvicensis*, *B. torta*, *B. valida*, and *B. wiebkei*), exclusive to the Hawaiian Islands that exhibit a wide range of morphological and ecological characteristics [12]. It is noteworthy that all of these variations have evolved from a common ancestral species through the process of adaptive radiation. The study examined an in-depth analysis of the flavonoid composition in Hawaiian *Bidens* species, identifying a diverse array of over 30 flavonoid compounds. The predominant constituents were chalcone and aurone derivatives, which constituted a significant portion of the overall flavonoid profile. Additionally, the study detected flavanones such as eriodictyol, three different flavones including apigenin, luteolin, and diosmetin, and two flavonols including kaempferol and quercetin (Figure 3), all in various glycosylated forms. Interestingly, two compounds, diosmetin and okanin 3,3′,4,4′-tetramethyl ether (Figure 3), were found exclusively in their aglycone forms. The glycosides identified in these compounds were primarily based on glucose, with a single exception being luteolin 7-*O*-glucuronide (Figure 3), being observed in several taxa. Notably, the study highlighted the unusual presence of mono-, di-, and tri-acetylated aurone and chalcone glycosides in the flavonoid profiles. However, the specific positions of acetylation were not determined. Aurone glycosides were derived from sulfuretin (3′,4′,6-trihydroxyaurone) and maritimetin (3′,4′,6,7-tetrahydroxyaurone) (Figure 3), and their aglycones, mono-, and di-glucosides were identified. In the case of di-glucosides, the nature of the linkage of the outer glucose moiety was not determined. Additionally, a tri-glucoside form of maritimetin was found in the flowers of *B. cervicata*. The diversity of chalcone derivatives was rooted in two parent compounds, butein (2′,3,3′,4-tetrahydroxychalcone) and okanin (2′,3,3′,4,4′-pentahydroxychalcone) (Figure 3). These compounds were found as aglycones in some plants, monoglucosides in most, and diglucosides in a few. Butein 4-methyl ether was identified in two individuals of *B. sandvicensis* subsp. *confusa*, and these same individuals also contained okanin 3,4,3′-trimethyl ether 4′-glucoside (Figure 3). Various di-, tri-, and tetra-*O*-methyl derivatives of okanin had been reported previously in *B. torta*. Kaempferol and quercetin glycosides were observed in nearly all the taxa studied, with just two exceptions. *B. sandvicensis* subsp. *confusa* lacked these compounds but featured the *O*-methylflavone diosmetin (only in aglycone form), which was not detected in other species. The second exception was *B. macrocarpa*, in which flavonols were replaced by flavones, namely apigenin and luteolin, present as 7-mono- and di-glucosides. Luteolin 7-*O*-glucuronide (Figure 3) was also found in *B. macrocarpa* and several other taxa, such as *B. amplectens*, *B. cervicata*, *B. mauiensis*, and *B. forbesii* subsp. *forbesii*, although it was notably absent in *B. forbesii* subsp. *kahiliensis*. Maritimetin, or its derivatives, were encountered in all the taxa studied. However, the occurrence of maritimetin derivatives varied across species, ranging from just the presence of the aglycone in *B. cosmoides* and *B. campylotheca* subsp. *campylotheca* to a comprehensive array of maritimetin derivatives in *B. micrantha* subsp. *micrantha*. Importantly, different subspecies of the same species exhibited distinct arrays of maritimetin derivatives.

In contrast, the presence of okanin, butein, and sulfuretin derivatives did not follow a discernible pattern, with some taxa lacking these compounds entirely, while others contained various combinations. Notably, these pigments were observed primarily in floral tissue, with limited presence in leaves. For instance, okanin, butein, and sulfuretin derivatives were exclusively seen in the flowers of *B. conjuncta*. Substantial differences were also observed among subspecies of the same species, with varying flavonoid profiles noted. For instance, *B. micrantha* subspp. *micrantha* and *B. ctenophylla* exhibited derivatives of all three of these anthochlors, while *B. micrantha* subsp. *kalealaha* lacked them entirely. Furthermore, significant differences were apparent among subspecies of other species as well. For example, the three subspecies of *B. campylotheca* differed concerning several compounds, with *B. campylotheca* subsp. *campylotheca* predominantly featuring anthochlor aglycones in floral tissue, while the other two subspecies had a typical array of maritimetin derivatives but lacked the aglycones or derivatives of okanin, butein, or sulfuretin. Additionally, *B. campylotheca* subsp. *waihoiensis* lacked eriodictyol 7-*O*-glucoside (Figure 3), which was present in the other two subspecies. Comparisons between *B. forbesii* subsp. *forbesii* and *B. forbesii* subsp. *kahiliensis*, as well as *B. menziesii* subsp. *menziesii* and *B. menziesii* subsp. *filiformis*, revealed that members of each pair exhibited similar anthochlor patterns but differed concerning their flavone and flavonol glycoside patterns. Lastly, differences between *B. sandvicensis* subsp. *sandvicensis* and *B. sandvicensis* subsp. *confusa* were observed, particularly regarding flavonol glycosides, the unique presence of diosmetin in the latter, and the specific nature of their respective anthochlor patterns [13].

### 2.3. Charpentiera obovata *Gaudich*.

*Charpentiera obovata* belongs to the Amaranthaceae family and holds significance in global floras except for frigid regions. The genus *Charpentiera* is endemic to the Hawaiian Islands and primarily consists of three species: *C. elliptica* (Hbd.) Heller, *C. obovata* Gaud., and *C. ovata* Gaud. [5,14]. Generally, *C. obovata* adopts the form of a small tree, ranging from 15 to 35 ft in height, and thrives best in arid habitats. Notably, this species displays variability primarily in leaf morphology [14]. Its distribution encompasses all the islands of the Hawaiian archipelago, thriving in wetlands and dry forests at elevations up to 4000 ft. During full bloom, it exhibits distinct features such as inconspicuous glabrous flowers arranged in large pendulous spikes and a trunk that often splits into multiple column-like buttresses. The wood of *C. obovata* is remarkably soft, fibrous, and exceptionally lightweight when dry, making it easily ignitable. This characteristic, coupled with the tree’s susceptibility to combustion, led to its utilization by indigenous Hawaiians for remarkable firework displays. The leaves can grow up to 12 cm wide and 30 cm long, and both the leaves and bark emit a mildly pungent fragrance when crushed [6,7]. The initial exploration of alkaloid content in Hawaiian plants has revealed the presence of alkaloids, 4-methoxycanthin-6-one (Figure 4) [14,15] in the roots and barks of *C. obovata*. However, a comprehensive review of the existing chemical literature indicates a lack of studies elucidating the molecular structure of these alkaloids.

### 2.4. Clermontia persicifolia *Gaudich*.

*Clermontia persicifolia* is a flowering plant belonging to the Campanulaceae family, specifically within the subfamily Lobelioideae. It is exclusively found on Oahu [16]. These are 2–6 m tall shrubs or trees, found either on land or as epiphytes. The leaves are 7–16 cm long and 1.5–4 cm wide, elliptical, or oblanceolate, with a glossy dark green upper surface. Inflorescences hold two to six flowers, with smooth (glabrous) peduncles 0.6 to 3 cm long, and pedicels 1.2–3 cm long [17]. In 1972, Saleh and Towers documented the presence of flavonoids from the leaves of an ethanolic extract obtained from *C. persicifolia*. The extract was initially fractionated using a polyamide column and subsequently subjected to additional separation on paper. Through this analytical procedure, the flavonoids, apigenin-7-glucoside, apigenin-7-rutinoside, luteolin-7-glucoside, and luteolin-7-rutinoside (Figure 5) were effectively identified and characterized [18].

### 2.5. Coprosma ernodeoides *A. Gray*

*Coprosma ernodeoides* (Nene bush), also known as ‘Aiakanene in Hawaiian name, is a sprawling perennial prostrate shrub native and endemic to Hawaii and belongs to the Rubiaceae family. This genus comprises approximately 110 species worldwide, primarily found in the Pacific Islands, with New Zealand hosting 55 *Coprosma* species. The plant features tiny leaves measuring 2–3 mm, densely covering branches that can extend up to three m in length. Throughout different times of the year, the bush produces shiny black berries that are edible and range from 5 to 8 mm in diameter [19]. Native Hawaiians historically used these berries as laxatives. *C. ernodeoides* bears succulent, purple fruits containing sizable central seeds. These fruits are rich in moisture but have relatively low levels of other substances. Presumably, these juicy fruits play a vital role in providing essential hydration on the arid mountain slopes range [20]. In 2015, Lang and research team conducted the initial studies on methanol extracts obtained from both leaves and berries of *C. ernodeoides*. The investigation led to the isolation of known iridoid glycosides including asperuloside, asperulosidic acid, deacetylasperulosidic acid, and scandoside (Figure 6) from the berries and leaves. These studies uncovered antioxidant properties using DPPH and FRAP assays [19].

### 2.6. Cuscuta sandwichiana *Choisy*

*Cuscuta sandwichiana* is a parasitic vine endemic to the Hawaiian Islands, and an exclusive member of Convolvulaceae family with a longstanding history of traditional medicinal use among local herbalists. It has been employed for treating lung cancers in the islands [21]. In Polynesia, herbal medicine practitioners occasionally utilize stem infusion to address various health conditions and ailments; nonetheless, it is important to note that as of now, there has been no scientific research conducted on its potential medicinal properties [22]. The stems are slender to medium-sized. Flowers are typically small, measuring 3–4 mm (occasionally up to 5 mm) and mostly have five parts. They are often glandular and arranged in clusters. The calyx surrounds the corolla and has triangular-ovate lobes with pointed or somewhat blunt tips. In some flowers, these lobes may form a slight ridge along the middle. The corolla starts as bell-shaped (campanulate) but becomes more globular as the capsule develops. It is thin, and the lobes are either ovate or somewhat triangular, particularly in older flowers [23,24].

In 1990, Jang’s initial research findings centered around the discovery of cytotoxic macrocyclic glycoresins extracted from the Hawaiian medicinal plant, *C. sandwichiana*. These glycoresins featured 11-hydroxy-tetradecanoic acid as their primary component, accompanied by a minor aglycon known as dihexadecanoic acid. Moreover, these compounds were found in conjunction with various sugars, including *L*-rhamnose, *D*-quinovose, *D*-fucose, and *D*-glucose. Additionally, a complex glycoside structure was identified, characterized as 11*R*-hydroxy-tetradecanoic acid 11-*O*-*α*-*L*-rhamnopyranosyl-(1-3)-*O*-*α*-*L*-rhamnopyranosyl-(1-2)-*O*-*β*-*D*-glucopyranosyl-(1-2)-*β*-*D*-fucopyranoside [21].

In 1996, Locher and colleagues investigated the antiviral activity of Hawaiian medicinal plants against human immunodeficiency virus type-1 (HIV-1). Among these plants, *C. sandwichiana* was examined. Methanol and aqueous extracts of *C. sandwichiana* stem were studied for their ability to selectively inhibit viral growth using the LAI (HTLV-III B) strain of HIV-1. The results indicated that the methanol extract had a 50% effective inhibitory concentration (EC_50_) of 26.8 ± 11.3 μg/mL, while the aqueous extract had an EC_50_ of 29.7 ± 6.3 μg/mL. The 50% cytotoxic concentration (CC_50_) for the methanol and aqueous extracts was greater than 223.1 ± 46.6 μg/mL and 124.6 ± 10.4 μg/mL, respectively. In terms of percent protection against viral cytopathic effects compared to control wells, the methanol extract showed 91% protection, while the aqueous extract exhibited 68% protection [25].

### 2.7. Dryopteris mauiensis *C. Chr.*

The genus *Dryopteris* is widely distributed, encompassing around 225 species. These species are primarily found in temperate forests and mountainous regions within tropical areas. In the Hawaiian Islands, an estimated eight to 17 species of *Dryopteris* exist, and all of them are unique to the islands. Among these, *Dryopteris mauiensis* is one of the endemic species found in the region [26]. *D. mauiensis* is a fern genus in the family Dryopteridaceae. These plants are sizable and thrive in terrestrial environments, typically found in moist forests across all the major Hawaiian Islands at elevations ranging from 945 to over 1700 m. They possess upright rhizomes with a diameter of 8 to 16 cm. The fronds of these plants are notably large, measuring between 75 and 260 cm in length. They have 12–17 pairs of pinnae, which are stalked and arranged alternately. The pinnules of these fronds are oblong-lanceolate to oblong-ovate in shape and can range from having stalks to being nearly sessile [26,27].

A comprehensive worldwide study delved into the phloroglucinol composition within the *Dryopteris* genus, leading to discussions on taxonomy that incorporated both morphological and chemical aspects. Interestingly, the Hawaiian species *D. mauiensis*, as reported by Widén in 2015, was found to either contain only minimal traces of phenolics (phloroglucinols) or, in some cases, to completely lack them [28].

### 2.8. Dubautia arborea *(A. Gray) D. D. Keck*

*Dubautia arborea*, a member of the Asteraceae family, is a substantial shrub to a small tree exclusively found on the Island of Hawaii. It thrives in scrublands and alpine desert environments, flourishing at elevations ranging from 2125 to 3000 m. *Dubautia*, together with *Argyroxiphium* and *Wilkesia*, forms what is commonly known as the silversword alliance [29,30]. Flavonoid exudates were extracted from leaf samples through brief rinses with dichloromethane, and the resulting mixture was separated using column and thin-layer chromatography (TLC) techniques. The structures of these compounds were elucidated using standard methods such as UV, PMR, and MS. Among the major compounds identified were apigenin (Figure 3), luteolin 4′-methyl ether (diosmetin) (Figure 3), luteolin 7-methyl ether, 6-methoxyluteolin, naringenin 7-methyl ether (sakuranetin), eriodictyol 7-methyl ether, and 5,4′-dihydroxy-6,7-dimethoxy-flavone (cirsimaritin) (Figure 7) as published by Bohm in 1999 [29].

### 2.9. Erythrina sandwicensis *O. Deg.*

*Erythrina sandwicensis*, commonly known as wiliwili and belonging to the Fabaceae family, holds a prominent position as an endemic species within Hawaiian tropical dry forests [31]. It can attain heights of 9–13 m, featuring a wide-spreading growth pattern. The leaves are arranged alternately, compound in nature, measuring 13–30 cm in length, and attached to a long and slender leafstalk. These leaves consist of three leaflets, with the one located farthest from the stem being larger than the other two. Each leaflet measures 4–10 cm in length and 6–15 cm in width [32].

During the years 1939 and 1940, Folkers and Koniuszy conducted a comprehensive investigation into the alkaloids content of various *Erythrina* species, including *E. sandwicensis*. Their research yielded the discovery of four previously unidentified alkaloids, namely erythramine, erysodine, erysopine, and erysovine, in addition to the presence of hypaphorine (Figure 8), all of which were successfully isolated from seed extracts of *E. sandwicensis*. Erythramine exhibited pronounced efficacy in inducing a paralysis resembling that caused by curare in frogs. This study contributed valuable insights into the chemical composition of *E. sandwicensis* and its alkaloid profile [33,34,35].

In 1980, Ingham documented the discovery of two novel phytoalexins found in the fungus-inoculated leaves of *E. sandwicensis*. These compounds were identified as (–)-6a*S*; 11a*S*-3,6a,9-trihydroxy-10-isopentenylpterocarpan (sandwicarpin) and (–)-6a*R*; 11a*R*-3-hydroxy-9-methoxy-10-isopentenylpterocarpan (sandwicensin). These newly identified compounds were found alongside several known pterocarpan derivatives (demethylmedicarpin, 3,6a,9-trihydroxy-pterocarpan, phaseollidin, and cristacarpin) and isoflavan derivatives (demethylvestitol and isovestitol) (Figure 9). The preparation and spectral characteristics (including UV and MS spectra) of 3-methoxy-9-hydroxy-10-isopentenylpterocarpan were also documented. Fungus-induced diffusates were analyzed and found to contain sandwicarpin and sandwicensin at concentrations ranging from 11–22 μg/mL and less than 0.3–1 μg/mL, respectively, based on a logarithmic scale (log ε = 3.78 at 286 nm for phaseollidin). Similarly, the concentrations of other *E. sandwicensis* isoflavonoids in the diffusates were as follows: demethylmedicarpin (0.5–1 μg/mL), phaseollidin (10–20 μg/mL), cristacarpin (4–10 μg/mL), 3,6a,9-trihydroxypterocarpan (1–3 μg/mL), demethylvestitol (9–24 μg/mL), and isovestitol (2–7 μg/mL) [36].

### 2.10. Gardenia brighamii *H. Mann*

*Gardenia brighamii*, commonly known as nanu or nau, is an evergreen shrub belonging to the Rubiaceae family. This species was designated as endangered by the federal authorities in 1985. With fewer than 20 remaining individuals in the wild, it faces an imminent risk of extinction. *G. brighamii*, once a significant component of lowland dryland forests across the primary Hawaiian Islands, is currently threatened by urbanization, invasive plant species, and the impact of grazing and browsing by domestic and feral animals [37]. Notably, *G. brighamii* stands out for its larger, fragrant flowers and glossy leaves. Its presence in the drier, lower regions of five of the larger Hawaiian Islands suggests that it may be among the most primitive species within its genus, primarily due to its modestly sized, slender calyx lobes [38].

In 2010, Kafua conducted a study to explore the antifungal properties of leaf extracts from *G. brighamii* against five fumonisin-producing *Fusarium* species. The primary goal was to identify potential active compounds. The investigation involved in vitro assessments using the microtitre dilution method and a direct bioassay on TLC plates. The findings from both methods provided strong evidence of the antifungal capabilities of the plant. Specifically, the acetone leaf extract exhibited a minimum inhibitory concentration (MIC) of 3.25 mg/mL and a minimum fungicidal concentration (MFC) of 6.5 mg/mL against *F. verticillioides* and *F. oxysporum*. Conversely, the methanol and dichloromethane extracts displayed higher MIC values against the tested fungi. Moreover, the methanolic extract, when applied to TLC plates, demonstrated significant inhibition against *F. verticillioides* and *F. proliferatum*. Subsequently, using chromatography technique, 42 major fractions were isolated, leading to the identification of a single pure compound. The determining structure of this compound has not been officially documented or reported [39].

### 2.11. Hesperomannia arborescens *A. Gray*

*Hesperomannia arborescens* is a member of the Asteraceae family and is exclusively found in the Hawaiian Islands. It presents as a perennial shrub or tree. The leaf blades have an oblanceolate to obovate shape, with the lower leaf surface displaying sparse pubescence, particularly along the lower 1/3–1/2 portion of the midrib on young leaves. In contrast, the upper leaf surface is smooth (glabrous). The petioles measure about 1/7–1/4 of the total leaf length, while the peduncles are 8–13 mm in length. The middle involucral bracts have a width of 4–5 cm, and during the flowering stage, the involucre takes on a dusty pink color [40].

In 1995, Bohm and Stuessy conducted a comprehensive investigation into the flavonoid profiles of several species belonging to the genera within the Barnadesioideae subfamily of Asteraceae, which included the examination of *H. arborescens* as a focal point. The experimental procedure involved the extraction and subsequent fractionation of plant material following established protocols. The isolation of individual compounds was successfully achieved through column chromatography employing Sephadex LH-20, with elution performed using methanol and water mixtures. Final purification steps were accomplished using preparative thin layer chromatography (TLC). The assignment of compound structures was based on a comprehensive analysis that included chromatographic behavior, color reactions with Naturstoff reagent, standard ultraviolet spectrophotometric methods, and comparative assessments with known compounds. The results of this study revealed the predominant presence of the common flavonols, namely kaempferol (Figure 3), accompanied by a minor trace of kaempferol 3-*O*-rutinoside (Figure 10) [41].

### 2.12. Hillebrandia sandwicensis *Oliv.*

The genus *Hillebrandia* is a group of plants that includes only one species, *Hillebrandia sandwicensis*. *H. sandwicensis* is a plant that is native to the Hawaiian Islands and is the only member of the Begoniaceae family that is found in this region. The plant was first described by Oliver in 1866 and was named after Dr. Wilhelm Hillebrand [41], a physician and botanist who specialized in Hawaiian flora. *H. sandwicensis* is similar in appearance to *Begonia*, but it differs in several ways, including having more numerous and highly differentiated sepals and petals, a semi-inferior and incompletely closed ovary, and fruits that dehisce between the styles. The plant flowers from February to June and then becomes dormant from late summer until January. *H. sandwicensis* is found on the islands of Kauai, Maui, and Molokai, but it is now thought to be extinct on Oahu. The plant is most abundant on Kauai and Maui, but it is becoming increasingly rare even on these islands. The species is restricted to wet ravines in the montane rain forest zone at altitudes ranging from 900–1800 m, which is a habitat similar to that of many *Begonia* and *Symbegonia* species. Despite the wider occurrence of suitable habitat, especially on Hawaii, the most recently formed of the Hawaiian Islands, historical and current records of *Hillebrandia* populations show the species to be rare and localized [41]. The study analyzed the fresh leaves of *H. sandwicensis* and 126 *Begonia* taxa for flavonoids. Out of these taxa, *H. sandwicensis* was analyzed for flavonoids for the first time. The study found that *H. sandwicensis* endemic to Hawaii had only quercetin 3-*O*-rutinoside (Figure 11) as a flavonoid [42].

### 2.13. Lobelia yuccoides *Hillebr.*

*Lobelia yuccoides*, commonly known as Panaunau (Hawaiian name), is one of the five endemic species within the *Lobelia* genus. It is characterized as a stately shrub, earning its specific name “yuccoides” due to its resemblance to smaller yucca plants. The plant features a straightforward upright trunk with a slender woody zone and an extensive pith. The trunk is closely adorned with rows of rhomboidal leaf scars, and it supports a cluster of leaves at its apex. These leaves exhibit a grayish hue on their undersides and are dark green on their upper surfaces. Notably, *L. yuccoides* follows a monocarpic life cycle, meaning it blooms only once during its entire lifespan. During its flowering phase, it produces a solitary flower spike that can reach up to three ft in length, bearing as many as 400 flowers. It shares striking similarities with some species found in the Abyssinian highlands, which thrive at elevations of up to 14,000 ft. *L. yuccoides* primarily inhabits the ridges and canyons of mountainous regions on the leeward side of its native habitat, typically occurring at elevations around 3000 ft [43].

In an investigative study involving the alkaloidal extract derived from the root and stem bark of *L. yuccoides*, two-dimensional thin-layer chromatography revealed the presence of 10–14 distinct alkaloids. Among these alkaloids, one was successfully identified as 8-phenylnorlobelol (Figure 12) [43].

### 2.14. Lysimachia Genus

*Lysimachia* stands as one of the most extensive genera within the Primulaceae family, encompassing approximately 180 species of perennial or annual herbs, shrubs, subshrubs, and plants with upright or sprawling growth habits. Within this genus, *Lysimachiopsis* members exhibit a woody, perennial, scandent, or upright shrub-like morphology. Distinctive features such as the woody growth habit, 5–10 merous perianths, and tetracolporate pollen are noteworthy within the context of the Primulaceae family, strongly suggesting that this subgenus likely originated from a monophyletic ancestor [44,45].

The primary objectives of a study revolved around characterizing the flavonoids found in the endemic Hawaiian species of *Lysimachia*. This investigation sought to assess the potential of these flavonoids as valuable taxonomic markers and to juxtapose these findings with existing data on other members of the *Lysimachia* genus. Additionally, the study aimed to document variations in flavonoid profiles within and among populations, with the ultimate goal of facilitating comparisons with other island ecosystems [45]. The principal compounds detected across all 12 examined species (*Lysimachia daphnoides* Hillebr., *L. filifolia* C. N. Forbes & Lydgate, *L. glutinosa* Rock, *L*. *hillebrandii* Hook. f. ex A. Gray, *L. iniki* K. L. Marr, *L. kalalauensis* Skottsb., *L. maxima* (R. Knuth) H. St. John, *Lysimachia ovoidea* H. St. John, *L. pendens* K. L. Marr, *L. remyi* Hillebr., *L. scopulensis* K. L. Marr, *L. waianaeensis* H. St. John) predominantly belonged to the category of flavonol glycosides, with kaempferol (Figure 3), quercetin (Figure 3), and isorhamnetin (Figure 13) as their foundational constituents. Among these glycosides, those primarily encountered were 3-*O*-diglycosides and 3-*O*-triglycosides. Notably, the sole flavonol monoglycoside identified was quercetin 3-*O*-glucoside (Figure 13). Two additional compounds, vitexin and isovitexin (Figure 13), were observed in three (*L. pendens*, *L. remyi*, and *L. waianaeensis*) and two (*L. maxima* and *L. waianaeensis*) species, respectively. Moreover, *C*-glycosyl flavone eriodictyol 7-*O*-glucoside (Figure 3) was concurrently present in five species, namely *L. daphnoides*, *L. kalalauensis*, *L. maxima*, *L. remyi*, and *L. waianaeensis*. Interestingly, among the various *Lysimachia* species investigated, substantial variability in flavonoid distribution was observed. To provide an overview of the data, the total frequency of each flavonoid’s occurrence in the entire dataset is also presented. For instance, quercetin 3-*O*-rutinoside (Figure 11) was the sole compound present in all taxa the subsequent most prevalent compound was quercetin triglycoside [45].

### 2.15. Melicope barbigera *A. Gray* (syn. Pelea barbigera *Hillebr.*)

The genus *Melicope* (Rutaceae) encompasses approximately 235 species of shrubs and trees, with a distribution range spanning Southeast Asia and Australasia. This distribution extends westward to include the Mascarene Islands and Madagascar, and to the east, it encompasses most of the Pacific archipelagos. Among these species, there are 54 that are specifically endemic to the Hawaiian Islands, with 41 of them being characterized as single-island endemics [46]. In *M. barbigera*, a distinguishing characteristic is that the midrib of the leaf is typically smooth (glabrous) or nearly smooth, except for the presence of long-villous hairs along its sides [47].

In 2021, a research team led by Le conducted a study on the dichloromethane extract derived from the leaves of *M. barbigera*, an endemic plant species exclusively found on the Hawaiian island of Kauai. This investigation yielded the isolation of two acetophenones, named melibarbinons A and B (Figure 14), along with three 2*H*-benzopyranes, or chromenes, identified as melibarbichromens A and B, and alloevodionol (Figure 14). Additionally, isomeric melifolione compounds referred to as melifoliones A and B (Figure 14) were also successfully isolated. The structural elucidation of these isolated compounds, obtained from the dichloromethane extract following meticulous purification via chromatographic techniques, was rigorously achieved through a comprehensive suite of spectroscopic analyses. These analytical methodologies encompassed both 1D and 2D nuclear magnetic resonance (NMR) spectroscopy, as well as high-resolution electrospray ionization mass spectrometry (HRESIMS). Furthermore, the absolute configuration of these compounds was determined utilizing the modified Mosher’s method. Subsequently, all of the isolated compounds underwent evaluation for their cytotoxic activities against the A2780 human ovarian cancer cell line. Notably, melibarbinon B and melibarbichromen B exhibited cytotoxic activities, displaying IC_50_ values of 30.0 µM and 75.7 µM, respectively, in a nuclear shrinkage cytotoxicity assay and cisplatin was used as a positive control in this study [48].

### 2.16. Phyllanthus distichus *Hook. & Arn.*

*Phyllanthus distichus*, a member of the Phyllanthaceae family, is endemic to the Hawaiian Islands. It is typically encountered sporadically being locally abundant within mesic forests, often inhabiting steep slopes or ridge summits. On occasion, it can also be found in dry shrubland environments, specifically at elevations ranging from 60–950 m. Its distribution encompasses several Hawaiian Islands, including Kauai, Oahu, Molokai, Lanai, and the western regions of Maui, with occasional sightings on the eastern part of Maui, although it is rare in that locale [49]. These plants exhibit a growth form ranging from large shrubs to trees, with heights spanning from 0.9–5 m. The petioles of the leaves can extend up to 4 mm in length, while the leaf blades themselves measure between 7 and 80 mm in length and 5 and 32 mm in width. In the staminate (male) reproductive structures, the pedicels are typically 1.5–3 mm long [50].

In 2019, Win and colleagues conducted an assessment of the in vitro antioxidant, antiglycation, and antimicrobial properties of various indigenous medicinal plants found in Myanmar. Among these plants, *P. distichus* fruit extract demonstrated notable potency. Specifically, it exhibited a broad-spectrum antimicrobial effect against both Gram-positive and Gram-negative bacteria. The fruit extract of *P. distichus* was particularly remarkable, displaying robust inhibition of microbial growth across all tested microorganisms, including *S. aureus*, *B. cereus*, *E. coli*, *E. faecalis*, and *C. albicans*. The inhibition zone diameters for these microorganisms were measured at 17, 20, 21, 23, and 18 mm, respectively. Furthermore, it is worth noting that the antimicrobial activity of the *P. distichus* fruit extract was comparable to that of the standard antibiotic (chloramphenicol). The inhibition zone diameters achieved by chloramphenicol were 27, 24, 24, 24, and 30 mm against *S. aureus*, *B. cereus*, *E. coli*, *E. faecalis*, and *C. albicans*, respectively. The standard antibiotic chloramphenicol was effective against all the tested bacteria, while the 70% ethanol, used as the solvent for dissolving the plant extracts, had no impact on bacterial growth [51].

Later, a research study conducted by Thu and the research team centered around the utilization of *P. distichus* fruit for the purpose of wine production. Initially, a comprehensive assessment of the phytochemical compounds present in *P. distichus* fruits was conducted through phytochemical tests. The findings revealed that the fruits of *P. distichus* were rich in various phytochemicals, including alkaloids, glycosides, phenolic compounds, reducing sugars, saponins, polyphenols, tannins, flavonoids, and terpenes. Subsequently, two distinct types of *P. distichus* wine were crafted: one solely composed of *P. distichus* (referred to as X-type), and the other incorporating yeast in the fermentation process (referred to as Y-type). In both winemaking procedures, sugar concentrations were strategically employed. Following a maturation period of four months, the physicochemical properties of the resulting wines were meticulously analyzed. The pH levels of the X-type and Y-type wines were observed to be 3.69 and 3.67, respectively, indicating that both wine variants exhibited an acidic profile. Furthermore, the total dissolved solids (TDS) content in the X-type wine was measured at 1010 mg/L, while the Y-type wine contained 863 mg/L of TDS. Additionally, the sugar content in the X-type wine was determined to be 6.75 g/L, while the Y-type wine had a sugar content of 7.16 g/L. As part of the study, the antibacterial properties of the prepared wines were evaluated after four months of aging, employing the agar well diffusion method, chloramphenicol was used as the positive control. The results revealed that the prepared wines exhibited no discernible antibacterial activity against the selected microorganisms. Furthermore, the antioxidant activities of the wines were assessed using DPPH (2,2-diphenyl-1-picrylhydrazyl) assays and ascorbic acid was used as a positive control. The IC_50_ values, representing the concentration required to scavenge 50% of the DPPH radicals, were determined to be 56.65 μg/mL for the X-type wine and 57.16 μg/mL for the Y-type wine. These findings indicated the antioxidant potential of both types of star gooseberry wine [52].

In 2020, a research study conducted by Mohapatra, and colleagues delved into the assessment of the hypoglycemic and antidiabetic properties of *P. distichus* leaves in both normal and alloxan-induced diabetic rats. Employing a series of successive extractions, it was observed that the ethanolic extract of *P. distichus* leaves yielded a higher quantity when compared to the petroleum ether and chloroform extracts. Notably, the ethanolic extract demonstrated the most effective control of blood sugar levels in hyperglycemic Wistar rats, surpassing the performance of other experimental extracts. Furthermore, this particular extract exhibited significant reductions in blood sugar levels in normal animals as well. The investigation extended to the examination of the effects of alcoholic extracts of *P. distichus* leaves in normoglycemic and hyperglycemic animal models via oral administration. A crucial aspect of the study was the assessment of toxicity, which conclusively established the safety of alcoholic extracts from *P. distichus* leaves, even at higher doses of 3000 mg/kg body weight. This outcome underscored the absence of adverse effects on normal physiological and behavioral parameters. Moreover, the administration of alcoholic extracts from *P. distichus* leaves led to a substantial reduction in elevated glucose levels in alloxan-induced diabetic rats, affirming its efficacy as an antidiabetic agent. Remarkably, it also contributed to lowering normal glucose levels, indicative of its hypoglycemic properties. Additionally, in alloxan-administered rats, levels of whole protein, whole cholesterol, and enzyme activity for alanine amino transferase (ALAT), alkaline phosphate (ALP), and asparatate aminotransferase (ASAT) were significantly higher compared to normal rats. However, treatment with the test extract notably mitigated these elevated levels. Collectively, the findings from this investigation strongly suggest that alcoholic extracts derived from *P. distichus* leaves hold promise in the realm of diabetes mellitus therapy, serving as potential antidiabetic and hypoglycemic agents. The observed hypoglycemic and antidiabetic effects of the test extract may be attributed to its influence on glycogenesis, glycogenolysis, and metabolic activities, possibly mediated by one or more of its constituent compounds. It is worth noting that the ethanolic extract of *P. distichus* leaves exhibits a beneficial role in reducing blood sugar concentrations and managing various diabetic complications, alluding to its therapeutic potential [53].

### 2.17. Pipturus albidus *A. Gray ex H. Mann*

*Pipturus albidus* or mamaki (Hawaiian name) is a native and endemic plant in Hawaii which belongs to the nettle family (Urticaceae). It is a small tree or shrub that can reach a height of up to 10 m (about 33 ft) and is known for its distinctive heart-shaped leaves and white to pinkish flowers. The plant is unique to the Hawaiian Islands and holds cultural significance in traditional Hawaiian medicine and herbal practices. *P. albidus* has been traditionally used by the native Hawaiians for various medicinal purposes, including general debility, aid for the expectant mother, blood purifier, mild laxative [54,55,56,57,58]. *P. albidus*, a culturally significant plant in Hawaii, has been traditionally used for woodworking, textile production, and medicine. It was employed to make tools and durable bark cloth known as “Kapa”. *P. albidus* has a history of medicinal use, with leaves and fruits used to treat various ailments [57]. It is often brewed into an herbal tea [54,55,56,57,58,59,60,61], which is believed to have calming and healing properties as well as to treat various ailments, including stomach and liver problems [55]. In addition, the consumption of *P. albidus* teas was believed to have potential health benefits, such as reducing cholesterol levels, regulating blood glucose levels, and lowering blood pressure [57,61]. Beyond its medicinal uses, *P. albidus* leaves are also consumed as a beverage by many locals and visitors to Hawaii due to their pleasant taste and potential health benefits [57,60]. As with any herbal remedy, it is essential to exercise caution and seek professional advice before using *P. albidus* or any other plant for medicinal purposes.

In 1996, Locher conducted a study revealing that extracts from *P. albidus* exhibited significant and selective antiviral activity against herpes simplex virus 1 and 2 as well as vesicular stomatitis virus. Additionally, the extracts demonstrated growth inhibition of *Staphylococcus aureus* and *Streptococcus pyogenes*, two pathogenic bacteria. However, the antifungal activity of *P. albidus* extracts was observed to be comparatively limited [25]. Five distinct water extracts derived from *P. albidus* leaves demonstrated effective inhibition of HIV growth. The EC_50_ concentrations of these extracts ranged from 3.81 μg/mL to 29.6 μg/mL. Notably, four out of the five extracts provided complete protection (100%) against viral cytopathic effect, and the remaining extract offered 82% cell protection. Interestingly, the growth inhibition effect was not limited to water extracts or leaf extracts alone, as both water and methanol extracts obtained from the bark and leaves also exhibited significant inhibition of HIV growth [57].

Investigation of the nutritional values and mineral solubilities in *P. albidus* leaves and teas, compared them with other commercial teas. Seasonal variations were observed in macronutrients and minerals in both leaves. *P. albidus* exhibited higher potassium, calcium, magnesium, zinc, iron, and copper content than other commercial tea leaves. The phenolic acids (+)-catechins and rutin were identified in *P. albidus* leaves, with higher concentrations than certain tea leaves. *P. albidus* teas displayed relatively lower total antioxidant activity compared to green teas and Lipton teas, based on the photochemiluminescence method [54,61]. This study identified and quantified three phenolic acids, (+)-catechins, chlorogenic acid, and rutin, in *P. albidus* leaves using the liquid chromatograph-mass spectrometer technique. The concentrations of (+)-catechins, chlorogenic acid, and rutin varied from 1.1–5.0 mg/g in *P. albidus* leaves. Total antioxidant activity (TAA) in *P. albidus* teas brewed for 30 min ranged from 238–259 mg of ascorbic acid (AA)/g of tea, while longer brewing and storage resulted in decreasing TAA levels. Compared to other popular teas, *P. albidus* teas exhibited relatively low TAA levels but showed higher concentrations of (+)-catechins and rutin [55]. The antioxidant evaluation (the Ferric reducing antioxidant power (FRAP) assay) of *P. albidus* tea revealed a remarkable antioxidant activity of the FRAP value the FRAP value of 40.0 µM for a 1 µg *P. albidus*, surpassing other tested plants, and catechin was used as a positive control.

In anticancer and chemopreventive studies, *P. albidus* tea demonstrated greater effectiveness compared to green tea. The room temperature and boiling water extracts from dehydrated *P. albidus* tea powder exhibited the highest percentage inhibitions in the NF-κB assay (ranging from 73.7–75.0%), while freeze-dried *P. albidus* leaf powders brewed at room temperature showed the highest inhibitory activity in the nitrite assay and *N*-tosyl-*L*-phenylalanine chloromethyl ketone (TPCK) (5.1 ± 0.8 µM) was used as a positive control in the NF-kB assay. Furthermore, the 100% ethanol extract from dehydrated *P. albidus* tea leaves displayed the strongest cytotoxicity against breast cancer cells (71.3% survival rate) and staurosporine (30.8–68.3 nM) was used as a positive control in the cytotoxicity assay [62].

In 2023, Koher and a group of researchers undertook an extensive exploration into *P. albidus*, encompassing its ethnomedicinal legacy, research concerning its impact on human health, and an in-depth analysis of its phytochemical characteristics. *P. albidus* occupies a significant place in the cultural heritage of Native Hawaiian communities. Over centuries, it has been deeply integrated into traditional practices owing to its perceived medicinal properties and diverse material applications. *P. albidus* has been linked to a range of potential health benefits, encompassing antiviral, antifungal, antimicrobial, and anti-inflammatory attributes. The paucity of research data and the substantial gaps in the existing literature regarding *P. albidus* create challenges in comprehensively understanding its mechanisms of action. However, this dearth of information also opens up numerous opportunities for further exploration and inquiry. Viewed from cultural, medicinal, and ecological perspectives, it is evident that *P. albidus* holds immense significance and warrants in-depth investigation. Through the promotion of *P. albidus*, this review aspires to raise awareness about this remarkable plant, with the hope of fostering positive economic and medicinal outcomes that can contribute to the well-being of indigenous communities and the broader Hawaiian Islands [57].

Meesakul and colleagues provided a comprehensive summary of *P. albidus*, highlighting its notable antibacterial and antiviral properties. *P. albidus* has a longstanding history of traditional medicinal use, encompassing a range of therapeutic benefits. These include its antioxidative effects, mild natural laxative properties, anti-allergic attributes, support for cardiovascular and liver health, and its potential to reduce stress levels. Moreover, the leaves of *P. albidus* find utility in the preparation of herbal tea, which is believed to possess healing properties. The customary method involves steeping these leaves in water, whether they are fresh or dried. It is advisable to avoid boiling, as it may diminish the effectiveness of the tea. Historically, *P. albidus* tea has been employed to address various health concerns, such as stress, anxiety, allergies, and the regulation of blood pressure and cholesterol levels. Beyond its medicinal applications, the bark of *P. albidus* serves as a substitute for wauke (paper mulberry) in the creation of kapa or bark cloth, while its roots are harnessed for natural dyeing purposes. Lastly, the fruit of *P. albidus* is situated beneath the leaves of the plant [58].

### 2.18. Platydesma campanulatum *H. Mann* or P. spathulatum *(A. Gray) Skottsb.* or P. campanulata *H. Mann (syn.* Melicope spathulata *A. Gray)*


*Platydesma*, a genus belonging to the Rutaceae family, is exclusively found as endemic plants in the Hawaiian Islands and comprises four species. One of them, *Platydesma campanulata* [63,64], thrives at elevations ranging from 1500–5000 ft on the four largest islands of the archipelago. *P. campanulata* is characterized as a shrub or small tree with large leaves measuring approximately 20 × 50 cm^2^. Upon crushing the leaves, essential oils are released, while breaking the bark and wood emits a distinctive semeniferous odor. In the taxonomic classification, *Platydesma* is positioned among the *Xanthoxylae* of the *Rutoideae* subfamily, situated between the Mexican genus *Choisya* and the New Caledonian genus *Dutaillyea* [63,64]. However, Stone suggests that the closest relative of this genus is the Australian genus *Medicosma*. Previous surveys have indicated the presence of alkaloids in *P. campanulata*, thereby motivating a comprehensive study of this member of the endemic genus. The initial investigation involved collecting plant material in the Pupukea area of the Koolau range on Oahu, with taxonomic identification [63,64]. Subsequent collections for the main part of the study were obtained from the Kokee region on Kauai and the Kohala mountains on the Island of Hawaii [63,64]. The roots and barks of *P. campanulata* from each collection were processed separately, with the samples from Kauai and Hawaii kept distinct. The leaves of *P. campanulata* from all collections were processed together. This approach resulted in the isolation of alkaloids and furoquinolines: evolitrine, kokusaginine, and 6-methoxydictamnine from the roots and barks of the Kauai and Hawaii specimens, and platydesmine and 1,2-dimethyl-4-quinolone (Figure 15) from the roots and barks of the Hawaii specimens. The combined leaves yielded kokusaginine, 1,2-dimethyl-4-quinolone, and pilokeanine (Figure 15) in the form of a picrate compound [64].

### 2.19. Psychotria hawaiiensis *(A. Gray) Fosberg*

*Psychotria hawaiiensis*, a member of the Rubiaceae family, comprises a population of trees and shrubs thriving in the understory of montane wet forests. This species is endemic to the Hawaiian Islands. Notably, these trees can reach a height of up to 12 m and have twigs that vary in color, ranging from grey, red, to yellow brown. Their stipules are broadly ovate to obovate and can grow up to 8 mm in length. The leaves are supported by petioles that measure between 0.5 and 4.7 cm. The fruit typically ranges from 6 to 10 mm in length, although it is usually less than 8 mm. The inflorescence typically exhibits three to four orders of branching [65]. According to information obtained through personal communication*, P. hawaiiensis* has a historical tradition of use for treating cuts and wounds [25,54].

In 1995, Locher and colleagues embarked on a study in which they specifically selected plants with a documented history of utilization in Polynesian traditional medicine for addressing infectious diseases. Their research aimed to investigate the in vitro anti-viral, antifungal, and antibacterial activities of these selected plants, which included bark and leaf extracts from *P. hawaiiensis*. Notably, the findings revealed that aqueous and methanol extracts of *P. hawaiiensis* leaf and bark exhibited discernable antiviral effects against herpes simplex virus 1 and 2A. The antiviral activity of *P. hawaiiensis* was assessed by determining the inhibitory concentrations of aqueous leaf extracts, methanol leaf extracts, and methanol bark extracts, which were found to be 250 µg/mL, 250 µg/mL, and 125 µg/mL, respectively. Furthermore, both acetonitrile bark and leaf extracts of *P. hawaiiensis* exhibited inhibitory effects on the growth of various fungal strains, including *Microsporum canis*, *Trichophyton rubrum*, and *Epidermophyton floccosum*. The results indicated that both extracts inhibited the growth of *M. canis* and *T. rubrum* at the concentration of 1000 µg/mL. An acetonitrile extract of *P. hawaiiensis* bark demonstrated growth inhibition against *E. floccosum* at the concentration of 20 µg/mL, whereas an acetonitrile leaf extract exhibited growth inhibition at the concentration of 125 µg/mL. Importantly, this study provided empirical validation for some of the ethnobotanical accounts concerning the therapeutic efficacy of Hawaiian medicinal plants against infectious agents, employing in vitro biological assays [54].

In 1996, Locher and a collaborator conducted a study to investigate the antiviral activity of Hawaiian medicinal plants against human immunodeficiency virus type-1 (HIV-1), including the examination of *P. hawaiiensis*. *P. hawaiiensis* underwent testing to assess its capacity for selective inhibition of viral growth using the LAI (HTLV-III B) strain, and it demonstrated notable antiviral activity. Specifically, the aqueous, methanol, and acetonitrile bark extracts of *P. hawaiiensis* exhibited a half-maximal effective concentration (EC_50_) at 9.7 ± 6.2, 161.3 ± 68.2, and 101.6 ± 95 µg/mL, respectively. The corresponding 50% cytotoxic concentrations (CC_50_) for these extracts were 223.3 ± 4.6, 229.5 ± 29.0, and 154.0 ± 20.8 µg/mL, respectively. Regarding the leaf extracts of *P. hawaiiensis*, the aqueous extract displayed EC_50_ and CC_50_ both exceeding 250.0 µg/mL. On the other hand, the methanol leaf extract of *P. hawaiiensis* exhibited EC_50_ and CC_50_ both at 247.4 ± 3.7 µg/mL. These findings underscore the antiviral potential of *P. hawaiiensis*, particularly in the context of its inhibitory effects on viral growth [25].

### 2.20. Rauvolfia sandwicensis *A. DC.*

*Rauvolfia sandwicensis*, commonly referred to as Hao (Hawaiian name), is a tree belonging to the Apocynaceae family, situated in mesic forests, and remaining low-elevation dry forests or shrublands across the primary Hawaiian Islands. Additionally, this tree has been documented in lowland lava flow areas on the islands of Maui and Hawaii. [66]. *R. sandwicensis* stood as the dominant arboreal species within an enclave encircled by recent lava flows. These mature trees boasted an average height of 10 m. Notably, one particularly sizable tree, positioned at a precipice, was measured at its base, and yielded a diameter measurement of 29.0 cm. The majority of *R. sandwicensis* trees, encompassing approximately 96% of the population, display flowering activity in the month of June. However, there was a limited presence of observed fruit-bearing trees [67].

In 1957, Gorman and colleagues conducted a study on the alkaloid composition of Hawaiian *Rauworfia* species, with a particular focus on *R. sandwicensis*. The study found the presence of well-known alkaloids such as tetraphylline and tetraphyllicine (Figure 16). Furthermore, they found larger quantities of an isomer of indole hemi-acetal, ajmaline, which they named sandwicine (Figure 16). The intriguing aspect of sandwicine was its ability to yield distinct hydro derivatives, which could be transformed into a common indole hemi-acetal through lead tetra-acetate oxidation. This observation led to the conclusion that sandwicine represents the C-17 epimer of ajmaline. Additionally, the researchers determined that dihydrosandwicine was identical to tetrahydroajmalidine, the product obtained through sodium borohydride reduction of ajmalidine. This finding provided compelling evidence that ajmalidine (Figure 16) is, in fact, 17-dehydroajmaline. The study by Gorman and Neuss thus contributed valuable insights into the alkaloid composition and structural relationships among these compounds within the Hawaiian *Rauworfia* species [68,69].

### 2.21. Santalum paniculatum *Hook. & Arn.*

*Santalum paniculatum* known as sandalwood (Santalaceae), an endemic plant found only on Hawaii Island, thrives in dry forest areas on lava substrates or cinder cones, spanning elevations from 450–2550 m. This shrub or tree typically stands 3–10 m tall with a canopy diameter of 3–7 m, but it can reach heights of up to 15–20 m in sheltered environments. It starts flowering at around 3–4 years of age, but substantial flowering and fruiting may take 7–10 years. Flowering and fruiting occur throughout the year, often with two peaks. The leaves are 2.5–8 cm long and 2–4.5 cm wide, with glossy upper sides and sometimes pale lower sides. They may have glaucous surfaces of varying colors, such as yellowish-orange, bluish, or olive green. Mature fruits are purple to black drupes measuring 10–12 mm, featuring a distinct apical ring. In Hawaii, these plants are primarily cultivated within their natural range for economic and conservation purposes [70].

*S. paniculatum*, known for its commercial-grade oil production, is the source of essential oils. The oil is extracted from *S. paniculatum* trees found on Hawaii Island. The raw material for distillation currently comes from trees that are either dead and still standing or affected by heartwood rot. These trees are harvested from land that has been utilized for cattle grazing for the past 150 years [71].

In 2014, Braun and colleagues conducted an analytical study on Hawaiian sandalwood oil, which is derived from the wood of *S. paniculatum* sourced from Hawaii Island, often referred to as “The Big Island.” The analysis employed gas chromatography (GC) and gas chromatography–mass spectrometry (GC-MS) techniques to investigate four distinct commercial grades of this oil. The predominant constituents identified within these oils were (*Z*)-α-santalol, representing a range of 34.5–40.4% of the composition, and (*Z*)-*β*-santalol, comprising between 11.0% and 16.2%. Additionally, a sensory evaluation was conducted to assess the aroma of these oils in comparison to East Indian sandalwood oil. Furthermore, the chemical composition of Hawaiian sandalwood oil was subjected to comparative analysis with that of four distinct *Santalum* species originating from East India, New Caledonia, Eastern Polynesia, and Australia, respectively [72].

In 2016, Chavan conducted a research investigation into the antimicrobial properties of diverse medicinal plant extracts, notably *S. paniculatum* against specific microorganisms. Specifically, the study aimed to evaluate the effectiveness of these plant extracts, derived from their leaves, against a panel of microorganisms, including *Escherichia coli*, *Salmonella typhi*, *Proteus vulgaris*, and *Staphylococcus aureus*. The results of the study revealed that extracts from the leaves of *S. paniculatum* demonstrated antimicrobial activity, particularly against *Staphylococcus aureus*. The observed inhibition zone for this activity ranged between 6.5 and 8 mm [73].

In 2016, Ofori and colleagues conducted a study to explore the potential of high-performance thin-layer chromatography (HPTLC) for characterizing the essential oils of different *Santalum* species, especially, *S. paniculatum*. The research aimed to document the variations present in the essential oils of this species and to employ HPTLC band and peak intensity profiles of mixed oil samples to create more comprehensive representations. Six distinct oil samples of *S. paniculatum*, sourced from Hawaii, were subjected to separation via HPTLC. All six samples exhibited nine prominent bands at various retention factors (R*_F_*) when observed at 366 nm after derivatization, with colors ranging from yellow to orange. These bands were identified at R*_F_* values of 0.07 (yellow), 0.08, 0.10, 0.17, 0.39 (orange), 0.61, 0.63, 0.65, and 0.80. Notably, there was variability in both the intensity and band profile among these S*. paniculatum* oil samples. Furthermore, all the *S. paniculatum* oil samples featured a strong band corresponding to (*Z*)-α-santalol. Subsequent GC-MS analysis of the *S. paniculatum* oils confirmed the presence of (*Z*)-*α*-santalol in these samples, with percentages ranging from 31.2–42.9%. To generate a more comprehensive HPTLC profile representative of *S. paniculatum* oils, a pooled sample comprising eight oils was separated using HPTLC. This pooled sample’s HPTLC fingerprinting effectively captured the variations observed in individual oils. Further distinctions were observed between the oils, particularly when the derivatization plates were examined under a wavelength of 366 nm using white light. The band profiles of pooled oils from *S. paniculatum*, separated via HPTLC, revealed that *S. paniculatum* exhibited similar band profiles, characterized by a prominent band for (*Z*)-α-santalol and a yellow band at R*_F_* 0.07. Under a wavelength of 254 nm before derivatization, the band profiles of the four pooled sandalwood oils separated by HPTLC could be categorized into two groups. Pooled oils from *S. paniculatum* displayed comparable band profiles, featuring a common band at R*_F_* 0.60 [74].

In 2022, Wagan and research colleagues embarked on a comprehensive investigation involving the essential oils extracted from *S. paniculatum*. Their primary objective was to assess the efficacy of these essential oils in repelling the adult form of *Tribolium castaneum*, commonly known as the red flour beetle. Additionally, they sought to determine the essential oils potential in inducing larval mortality through fumigation and their capacity to impede the hatching of eggs laid by this pest. The findings of their study revealed that among the tested essential oils, the one derived from *S. paniculatum* exhibited the most notable pharmacological effectiveness against the red flour beetle. Particularly noteworthy was the significant reduction in egg hatchability, with *S. paniculatum* demonstrating a substantial 76% reduction after a 15-day exposure period. The research revealed that the essential oil extracted from *S. paniculatum* exhibited the second highest degree of larval toxicity, while also demonstrating noteworthy efficacy in repelling adult beetles and reducing egg-hatch rates. These results suggest that essential oils, while not providing a complete solution, may hold promise in the management of red flour beetles. The application of GCMS allowed for the identification of chemical components, determination of their retention times, and calculation of yield percentages for the extract. The analysis of the *S. paniculatum* oil unveiled the presence of a total of 13 compounds including various terpenoids. Following the assessment of bioactivity in the various essential oils, each underwent thorough GC-MS analysis to elucidate the chemical constituents responsible for their effects [75].

### 2.22. Sophora chrysophylla *(Salisb.) Seem.*

*Sophora chrysophylla*, a member of the Leguminosae family, is the only species of *Sophora* found in Hawaii. *S. chrysophylla* is distributed across the main islands, thriving from sea level, where it takes the form of a shrub, to higher elevations of up to 3000 m, where it transforms into a tall tree reaching heights of 15 m. *S. chrysophylla* was first phytochemically assayed in 1976 by Kadooka et al., who reported that two previously unknown quinolizidine alkaloids such as mamanine and pohakuline (Figure 17) were extracted from the bark of *S. chrysophylla*. These alkaloids are unique as they are 1-hydroxymethylenequinolizidines. These compounds could potentially serve as intermediate stages in an undiscovered biogenetic pathway of *Sophora* alkaloids [76]. A novel lupin alkaloid was discovered in *S. chrysophylla* leaves, stems, and seeds [77]. Isoflavones and 6a-hydroxypterocarpan were isolated from the roots of *S. chrysophylla*. These compounds were identified as sophoraisoflavanone C–D, and (–)-tuberosin (Figure 17) [78].

### 2.23. Vaccinium *Genus*

The *Vaccinium* genus encompasses over 400 species of shrubs or small trees, both terrestrial and epiphytic, traditionally classified into 33 sections. Within this taxonomic framework, the section *Myrtillus* stands out with seven distinct species, namely *V. caespitosum*, *V. deliciosum*, *V. membranaceum*, *V. myrtillus*, *V. ovalifolium*, *V. parvifolium*, and *V. scoparium*. These species exhibit several shared morphological characteristics, most notably featuring solitary flowers in the lowermost leaf axils and pedicels seamlessly connected to the calyx tubes. Through a comprehensive morphological examination of section *Myrtillus*, employing phenetic analyses, researchers sought to elucidate its taxonomic relationships. Concurrently, phylogenetic investigations leveraging molecular data from nuclear ribosomal internal transcribed spacer (nrITS) and chloroplast genes matK and ndhF unveiled a noteworthy revelation. This molecular analysis identified a monophyletic group comprising three Hawaiian species (*V. dentatum*, *V. calycinum*, and *V. reticulatum*), which were previously classified within *Macropelma*. Intriguingly, this group was determined to be derived from within the larger section *Myrtillus*, signifying a shift in our understanding of their evolutionary relationships [79]. Within the *Vaccinium* genus, it has been documented in the scientific literature that there are solely two reported species, namely *V. calycinum* and *V. reticulatum*.

#### 2.23.1. *Vaccinium calycinum* Sm.

*Vaccinium calycinum*, a member of the Ericaceae family, is a stiffly upright shrub ranging from 1–5 m in height, often growing in small clusters with 2–30 stems. Its deciduous leaves are chartaceous and have an ovate to obovate shape, measuring 5–8 cm in length and 2–4 cm in width. The leaf blades are typically smooth (glabrous) and feature serrated margins. The pedicels, or flower stalks, are 1–3 cm long. The calyx, initially somewhat puberulent when young, consists of lobes that are deltoid in shape, leaf-like, and overlapping in bud. These lobes have a width of 2–3 mm at the base and are longer than the tube during flowering. The corolla can vary in color, appearing green, yellowish green, or reddish green, and it assumes an urceolate to cylindrical shape, measuring 9–12 mm in length. The berries produced by *V. calycinum* are bright red, but they lack a distinct flavor and typically measure between 9 and 15 mm in diameter. The seeds are small and numerous [80].

#### 2.23.2. *Vaccinium reticulatum* Sm.

*Vaccinium reticulatum*, an Ericaceae member, is a rhizomatous shrub with stiffly erect aerial shoots, typically reaching heights from 10–200 cm. The current season’s twigs may exhibit pubescence, occasionally appear glaucous, or show both glaucous and pubescent characteristics. Its persistent leaves are coriaceous, typically ovate to obovate (occasionally elliptic), measuring 1–3 cm in both length and width. These leaves may display pubescence, glaucousness, or a combination of both, with margins that can be either sharply serrated or entirely smooth. The leaves themselves may be flat or strongly revolute. Pedicels, which support the flowers, range from 1–3 cm in length. The calyx can be glabrous, glaucous, or pubescent when young, with lobes that are deltoid in shape. During flowering, these lobes may be shorter or occasionally as long as the tube. The corolla varies in color, appearing red, yellow, yellow with red stripes, or greenish yellow, and it assumes an urceolate to cylindrical shape, measuring 8–12 mm in length [80].

In 1994, Bohm and Abyazani conducted a study focused on the isolation and structural elucidation of flavonoids and tannins in two endemics Hawaiian *Vaccinium* species, *V. reticulatum* and *V. calycinum*. Both species were found to contain similar flavonoids, including quercetin-3-*O*-glucoside (Figure 13), isorhamnetin (Figure 13), quercetin (Figure 3), quercetin-3-*O*-galactoside, quercetin 3-*O*-methyl ether, neochlorogenic acid, and (–)-epicatechin (Figure 18). The tannins identified in these species consisted of procyanidin units with exclusively *cis* stereochemistry. Additionally, a substantial quantity of neochlorogenic acid, a derivative of caffeic acid, was detected. It is suggested that the phenolic chemistry observed in these species may have been preserved in the common ancestor of Hawaiian *Vaccinium* [81].

In 2013, Hummer and research team aimed to conduct a comprehensive evaluation of phytochemical compounds present in the fruits of two Hawaiian cranberry relatives, namely *V. reticulatum* and *V. calycinum*. Various analytical techniques, including normal-phase high-performance liquid chromatography (HPLC) coupled with fluorescence and electrospray ionization mass spectrometry (ESI-MS) were employed. The colorimetric 4-dimethylaminocinnamaldehyde (DMAC) assay to quantify the total content of pro-anthocyanidins (PACs) was also used. Additionally, a range of spectrophotometric tests and reverse-phase HPLC, equipped with photodiode array and refractive index detectors, were used to assess the presence of phenolic compounds, sugars, and organic acids. Antioxidant activities were determined using the oxygen radical absorbance capacity (ORAC) and Ferric reducing antioxidant power (FRAP) assays. The results of their study demonstrated that two *Vaccinium* species exhibited notably high antioxidant activities. For instance, the FRAP measurement for pressed fruit of *V. calycinum* yielded a value of 454.7 ± 90.2 µmol/L Trolox equivalents/kg. Comparative analysis revealed some distinctions between two *Vaccinium* species, such as lower levels of peonidin, quinic and citric acids, and an approximately 1:1 glucose/fructose ratio in two *Vaccinium* species. The FRAP assessment yielded a measurement of 454.7 ± 90.2 µmol/L Trolox equivalents/kg for pressed *V. calycinum* fruit. Hawaiian berries exhibited comparatively lower levels of peonidin, quinic acid, citric acid, and an approximately equal glucose/fructose ratio when contrasted with cranberries. Both Hawaiian *Vaccinium* species demonstrated notable quantities of pro-anthocyanidins (PACs), encompassing phenolics and a spectrum of PAC monomers, A- and B-type trimers, tetramers, and larger polymer compounds. In comparison to cranberries, *V. reticulatum* and *V. calycinum* showcased equivalent or higher PAC concentrations. Cranberries exhibited a higher proportion of A-type dimers in contrast to *V. reticulatum*, while *V. calycinum* did not distinguish between A- and B-type dimers. The total PAC content, as quantified by the DMAC method, in *V. calycinum* (24.3 ± 0.10 mg catechin equivalents/kg) was approximately twice that observed in cranberries. These findings collectively suggest that berries from *V. reticulatum* and *V. calycinum* hold substantial potential as a rich dietary source of PACs, rivalling or surpassing the levels found in cranberries. Consequently, these Hawaiian *Vaccinium* species have the potential to be considered as functional foods. Further exploration of the phytochemical profiles of other wild *Vaccinium* species is warranted to expand our understanding of their nutritional and health-promoting properties [82].

In 2022, Wu and their research team embarked on a study with the primary objective of assessing the antimicrobial properties of *V. calycinum*, a Hawaiian wild berry closely related to cranberries. Their investigation focused on the effectiveness of *V. calycinum* extracts against *Listeria monocytogenes* in both culture media and dairy products. Additionally, the study explored the impact of *V. calycinum* juice, particularly at sub-inhibitory concentrations, on various physicochemical properties, biofilm formation, and gene expression patterns of *L. monocytogenes*. The study revealed that the minimum inhibitory concentration of *V. calycinum* juice required to inhibit the growth of *L. monocytogenes* was determined to be 12.5%. Furthermore, when *L. monocytogenes* was exposed to a sub-inhibitory concentration of *V. calycinum* juice (6.25%), several notable effects were observed. These included a significant increase in auto-aggregation, a decrease in hydrophobicity, as well as reductions in swimming motility, swarming motility, and the ability to form biofilms by *L. monocytogenes*. Gene expression analysis also indicated significant downregulation of specific genes associated with motility (flaA), biofilm formation and resistance to disinfectants (sigB), invasion (iap), listeriolysin (hly), and phospholipase (plcA) in *L. monocytogenes* treated with the 6.25% *V. calycinum* juice. Moreover, the study demonstrated that *V. calycinum* supplementation significantly inhibited the growth of *L. monocytogenes* in both whole and skim milk, even when the milk was supplemented with 50% *V. calycinum* juice, regardless of its fat content. These findings underscore the potential of *V. calycinum* as a natural preservative and functional food ingredient with the capacity to mitigate the risk of *L. monocytogenes* infections [83].

Later, Liu and colleagues conducted a comprehensive investigation into the inhibitory effects of *V. calycinum* fractions on the growth of two significant bacterial pathogens, namely *L. monocytogenes* and *E. coli* (O157:H7). This study sought to elucidate which specific constituents within *V. calycinum* fruits contribute most significantly to its antimicrobial properties. To achieve this, the crude extract of *V. calycinum* fruits was fractionated into three distinct components: sugar plus organic acids (referred to as F1), non-anthocyanin phenolics (referred to as F2), and anthocyanins (referred to as F3). Subsequently, the researchers determined the minimum inhibitory concentration (MIC) and minimum bactericidal concentration (MBC) of these fractions against both *L. monocytogenes* and *E. coli* (O157:H7). The outcomes of this study revealed that F3 exhibited the highest concentrations of total phenolic compounds and anthocyanins among the fractions analyzed. Notably, all three fractions demonstrated significant reductions in bacterial growth compared to control conditions. Specifically, F1, at its native pH, exhibited identical MIC values (1.39/0.36 Bx/acid) and MBC values (5.55/1.06 Bx/acid) against both *L. monocytogenes* and *E. coli* (O157:H7). However, when F1 was neutralized, it lost its inhibitory effects on the growth of these pathogens. In addition, the MIC of F3 against *L. monocytogenes* was determined to be 13.69 mg/L cyanidin-3-glucoside equivalent, and this antimicrobial activity remained unaffected by neutralization. Importantly, it was observed that *L. monocytogenes* exhibited greater sensitivity to all fractions when compared to *E. coli* (O157:H7). These compelling findings collectively suggest that both phenolic compounds and organic acids present in *V. calycinum* fruits play pivotal roles in conferring their antimicrobial properties. Consequently, these fruits hold promise as natural food preservatives, which could be of significant benefit to the food industry [84].

### 2.24. Wikstroemia *Genus*

#### 2.24.1. *Wikstroemia monticola* Skottsb.

*Wikstroemia monticola*, commonly known as the montane false ohelo, is a diminutive arboreal species belonging to the Thymelaeaceae family. *W. monticola* is a frequently encountered diminutive shrub, characterized by small leaves, typically found in arid to moderately moist windward locations, where it thrives up to its highest elevational thresholds. Its distribution is limited to the Hawaiian archipelago, with the highest concentrations occurring on the island of Maui [85].

In 1983, Jolad and colleagues conducted a study focusing on the isolation of daphnane diterpenes from *W. monticola*. The primary objective of their investigation was to identify potential tumor-inhibitory compounds within *W. monticola*. The study involved the extraction of an ether fraction from *W. monticola*, followed by further fractionation, which ultimately yielded a mixture of compounds with high thin-layer chromatography (TLC) homogeneity. Subsequent separation of this mixture using reversed-phase high-performance liquid chromatography (HPLC) resulted in the isolation of six distinct daphnane diterpenes. These compounds were identified as wikstrotoxins A–D, huratoxin, and excoecariatoxin (Figure 19). The mixture, which exhibited a singular spot on the TLC analysis, displayed a notable level of activity, achieving a 158% response (T/C) at a dose of 75 μg/kg in the in vivo 3PS tumor model. Subsequently, both this mixture and the purified huratoxin have been subjected to further evaluation for their potential antitumor properties, with further investigations [86,87,88].

#### 2.24.2. *Wikstroemia uva-ursi* A. Gray

*Wikstroemia uva-ursi* (Thymelaeaceae), commonly referred to as ‘Ãkia in Hawaiian name, represents an endemic plant species that possesses versatile utility, serving as viable groundcover or a modest shrub. It is a compact groundcover species with oval, waxy green leaves arranged closely in an imbricate pattern. Its small yellowish-green flowers cluster together, followed by the development of orange-red berries. It typically grows horizontally, reaching a height of 1–3 ft but can extend laterally to widths of up to 10 ft in mature plants [89]. *W. uva-ursi* distinguishes itself by its noteworthy resilience to adverse conditions, such as drought, salinity, and high winds, while concurrently remaining relatively unscathed by significant pest infestations. The aesthetic appeal of *W. uva-ursi* is underscored by its vibrant red berries and aesthetically pleasing foliage, rendering it a potential candidate for ornamental horticulture. This botanical specimen holds cultural significance, as it finds application in crafting traditional Hawaiian headpiece through the utilization of its leaves, flowers, and fruits. Furthermore, its historical use in traditional Hawaiian medicine raises the prospect of pharmaceutical value. Notably, it was employed by ancient Hawaiians for medicinal purposes, and it served as an agent for fish anesthesia, facilitating their capture [90,91,92,93].

In 1997, Torrance and colleagues documented their discovery of the lignan named wikstromol (Figure 19) from *W. uva-ursi*. The research revealed that ethanol extracts derived from the entire *W. uva-ursi* plant exhibited notable antitumor efficacy when tested against the P-388 lymphocytic leukemia (3PS) experimental system. Among the constituents responsible for the observed antitumor activity in both plant species, wikstromol was identified as a potent component. Additionally, several other compounds, namely daphnoretin, pinoresinol, and syringaresinol (Figure 20), were identified as inactive in this context [87,94].

### 2.25. Wilkesia *Genus*

#### 2.25.1. *Wilkesia gymnoxiphium* A. Gray

*Wilkesia gymnoxiphium*, a member of the Asteraceae family, is characterized by its growth as a stalked rosette shrub, which can reach heights of up to 5 m when supported by woody stems. *W. gymnoxiphium* predominantly thrives in scrubland and open forest environments and exhibits a monocarpic life strategy which is Endemic plant to the Hawaiian Islands [95,96,97].

#### 2.25.2. *Wilkesia hobdyi* St. John 

*Wilkesia hobdyi* (Asteraceae), a species of limited abundance with approximately 350 known individuals, primarily inhabits dry ridges situated at elevations ranging from 275 to 400 m. This species typically exhibits branching characteristics starting from its lower portions and attains a maximum height of around 0.7 m. In contrast, *W. gymnoxiphium* assumes the form of an unbranched and monocarpic rosette plant, capable of reaching heights of up to 5 m, although it tends to be shorter in open environments. It is worth noting that the habitats favored by *W. hobdyi* tend to be somewhat drier compared to those preferred by *W. gymnoxiphium*. The latter species thrives in openings within both dry and mesic forest ecosystems, as well as on ridges. Importantly, both of these species are endemic to the island of Kauai, and their distinct habits set them apart from one another [98].

In 1990, Bohm and Fong conducted an investigation into the leaf exudate flavonoids of two *Wilkesia* species, *W. gymnoxiphium* and *W. hobdyi*. Their findings prompted attention to the issue of limited sampling within the Hawaiian Silversword Alliance (Madiinae). Of particular note was the observation that the flavonoid composition in *W. gymnoxiphium* and *W. hobdyi* exhibited a consistent qualitative profile, irrespective of whether one considered intra-population variations or comparisons across three distinct populations. Minor quantitative distinctions were the primary differentiating factors in the flavonoid content among the sampled specimens [99].

### 2.26. Zanthoxylum *Genus*

#### 2.26.1. *Zanthoxylum dipetalum* H. Mann

*Zanthoxylum dipetalum* (Rutaceae) is a small to medium-sized tree that can be distinguished from other endemic Hawaiian species by specific morphological characteristics. Notably, it possesses only two petals, resulting from the fusion of adjacent petals. Additionally, it exhibits a unique leaf structure, featuring a pair of modified basal leaflets attached to the lowest pair of regular leaflets, and its inflorescence pattern sets it apart from other species [100,101].

In 1975, Fish and colleagues conducted a study on the root of *Z. dipetalum*, identifying several bioactive compounds, including alkaloids (canthin-6-one, chelerythrine, nitidine, and tembetarine), pyranocoumarins (avicennol and xanthoxyletin), a triterpene (lupeol), and a flavonoid (hesperidin) (Figure 21). Furthermore, they discussed the mass spectrometry fragmentation pattern of avicennol and proposed a tentative structure for a third coumarin, designated as an analogous dipyrano coumarin (ZD/l). Notably, the root wood of the species and the stem bark of the γ variety contained most of these compounds, along with sitosterol (Figure 21), and, exclusively in the root wood, magnoflorine (Figure 21). The major constituent identified in the petroleum ether extract obtained from the root bark of *Z. dipetalum* was avicennol, a pyrano [2,3-f] coumarin. Thin-layer chromatography (TLC) analysis of this extract indicated the presence of minimal amounts of avicennol, xanthoxyletin, canthin-6-one, lupeol, sitosterol, and a compound referred to as an analogous dipyrano coumarin (ZD/l). Further analysis of the chloroform (CHCl_3_) extract revealed slight traces of canthin-6-one and chelerythrine chloride. In contrast, the methanol (MeOH) extract contained tembetarine and another quaternary alkaloid, likely to be magnoflorine. Additionally, hesperidin was isolated and identified from the MeOH extract. When the stem bark of *Z. dipetalum* was subjected to extraction, the petroleum ether concentrate was fractionated through column chromatography, yielding lupeol, xanthoxyletin, and sitosterol, all of which were positively identified by direct comparison with authentic reference materials. Minor quantities of canthin-6-one and avicennol were detected by TLC. Similarly, traces of chelerythrine and canthin-6-one were observed in the CHCl_3_ extract of the stem bark [101].

In 1976, Fish and his research collaborators published a significant discovery related to compounds isolated from the root bark of *Z. dipetalum*. They reported the isolation and structural confirmation of a novel dipyranocoumarin known as dipetalolactone. Its chemical structure was established through the synthesis of tetrahydrodipetalolactone. Furthermore, they identified another new pyranocoumarin, referred to as dipetaline, and tentatively assigned it the structure of 6-(3,3-dimethylallyl)-5-methoxy-22-dimethyl-2*H*-[1,2-b:3,4-b′] dipyran-8-one, based on proton magnetic resonance (PMR) analysis using the lanthanide shift reagent Eu(fod)_3_. They also noted the presence of a contaminating compound in the crystallization process of dipetalolactone, which was identified as another pyranocoumarin. This particular compound, termed dipetaline, was tentatively identified as an angular pyrano[2,3-f] coumarin [102].

In 1990, a study conducted by Arslanian and colleagues documented the isolation of three previously unreported natural acyl histamines (*N*-benzoylhistamine and *N*-(2-methoyxybenzoyl) histamine, and *N*-(2,3-dimethoyxbenzoyl) histamine) (Figure 21), in addition to one previously identified compound (cinnamoyl histamine) (Figure 21), from the leaves of various *Z. dipetalum* populations found across the Hawaiian Islands. Furthermore, they discovered the presence of the rare protopine-type alkaloid, thalictricine (Figure 21). Notably, the concentrations of acyl histamines and thalictricine were found to be remarkably consistent among populations originating from Hawaii and Kauai islands. However, it was observed that the composition of these compounds differed in a population originating from the island of Oahu [103].

#### 2.26.2. *Zanthoxylum hawaiiense* Hillebr.

*Zanthoxylum hawaiiense* is a small tree, typically ranging from 1–8 m in height. It falls within the Rutaceae family, sharing a similar size and growth pattern with previously discussed species. The leaves are pedately 3-foliolate, each leaflet attached to petioles, with the leaflets themselves affixed to nearly uniform-length non-articulate stalks. The follicles of this species exhibit a curved morphology and are characterized by a nearly smooth texture [100,104].

#### 2.26.3. *Zanthoxylum kauaense* A. Gray

*Zanthoxylum kauaense*, belonging to the Rutaceae family, is characterized by its stature as a tree of modest to moderate size, typically reaching heights ranging from 1–15 m. These trees feature trunks with diameters spanning 40–50 cm and display bark that ranges in color from brown to grayish brown. The newly developing growth on these trees is typically smooth or finely pubescent in texture. The leaves consist of three–seven (occasionally up to 11) leaflets, which are notably thick and possess a leathery texture, assuming either an ovate or elliptic shape. Additionally, this species produces flowers, typically numbering between 15 and 150 or more, which are arranged in axillary, open, cymose inflorescences, usually measuring 8–15 cm in length [80].

In 1992, Marr and colleagues conducted a study focusing on the volatile insecticidal compounds and chemical variability present in endemic Hawaiian *Zanthoxylum* species, specifically *Z. kauaense*, *Z. dipetalum*, and *Z. hawaiiense*. They employed a bioassay using fruit fly (*Dacus dorsalis* Hendel) eggs to assess the insecticidal properties of leaves and pericarps from 80 individual trees. Notably, the toxicity levels of extracts from different plants exhibited significant variation. Out of the 47 extracts from *Z. kauaense*, only 12 were found to prevent egg hatch, while one out of the 12 *Z. dipetalum* extracts exhibited similar activity. In contrast, none of the 21 *Z. hawaiiense* extracts displayed ovicidal properties. The researchers identified a total of 14 compounds in these extracts, encompassing aliphatic alcohols, ketones, mono- and sesquiterpenoids, and phenylpropanoids. The composition of these compounds varied across different levels of comparison, including among species, among and within populations, and between different plant tissues. In particular, *Z. kauaense* extracts that prevented egg hatch were primarily composed of 2-undecanone and 2-tridecanone as major constituents. Conversely, non-toxic extracts were dominated by limonene, caryophyllene, or an unknown compound. There was an exception where the pericarp extract from one tree contained limonene as the major compound but still exhibited toxicity with an LD_100_ of 25 mg/mL. *Z. hawaiiense* extracts shared some compounds with *Z. kauaense*, but certain constituents found in *Z. kauaense*, such as azulenes, *α*-guaiene, bulnesene, and longifolene, as minor components were absent. The proportions of terpenoids, aliphatic ketones, and unknown compounds varied among *Z. hawaiiense* extracts, with limonene and unknown being predominant in most cases. An interesting finding emerged from the analysis of *Z. dipetalum*, revealing two distinct chemotypes. Samples from Hawaii and Kauai contained 2-undecanone, 2-tridecanone, and caryophyllene as major compounds, while extracts from three trees sampled on Oahu lacked aliphatic ketones and were instead rich in anethole, estragole, and caryophyllene, emitting a distinct anise odor. One extract from the latter chemotype exhibited an LD_100_ of 100 mg/mL. The study employed extracts from 80 leaf and 30 pericarp samples obtained from 80 trees, evaluating the percentage of egg kill at different extraction ratios (10, 25, 50, and 100 mg tissue/mL hexane). LD_50_ values were not established due to the data exhibiting either 0% or 100% kill. Therefore, toxicity was reported as the LD_100_, which represented the lowest increment causing complete mortality. Non-toxic samples were those that did not prevent egg hatch at 100 mg/mL. Among *Z. kauaense* extracts, 10 out of 47 leaf and 12 out of 15 pericarp extracts exhibited toxicity, with pericarp extracts generally displaying higher activity. However, the level of toxicity varied significantly between individual trees. In contrast, all samples from *Z. hawaiiense* were non-toxic, and only one out of the 27 *Z. dipetalum* leaf extracts showed mild toxicity. Overall, this study emphasized the importance of considering chemical variations when evaluating the pesticidal activity of endemic Hawaiian *Zanthoxylum* species [105,106].

## 3. Conclusions

Hawaiian endemic plants offer a wealth of bioactive compounds with diverse and promising pharmacological properties (Appendix A). Alkaloids found in *A. glauca* var. *glauca*, such as protopine, allocryptopine, sanguinarine, berberine, and chelerythrine, exhibit antimicrobial, anti-inflammatory, and anticancer properties [9]. *Bidens* spp. are a versatile group, producing polyacetylenes and flavonoids known for their antitumor, antiviral, and anti-inflammatory activities [12,13]. *C. obovata* contains the alkaloid 4-methoxycanthin-6-one, showing potential as an antitumor and antimalarial agent [14,15]. Flavonoids in *C. persicifolia*, such as apigenin-7-glucoside, apigenin-7-rutinoside, luteolin-7-glucoside, and luteolin-7-rutinoside, contribute antioxidant and anti-inflammatory effects [18]. *C. ernodeoides* boasts iridoid glycosides like asperuloside and scandoside, which offer antioxidant, anti-inflammatory, and anticancer properties [19]. *C. sandwichiana* displays cytotoxic macrocyclic glycoresins and promising preliminary antiviral activity against HIV-1 [21,25]. *D. mauiensis* produces phloroglucinols with antioxidant and antimicrobial benefits [28]. *D. arborea* presents a spectrum of flavonoids, including apigenin, diosmetin, sakuranetin, and cirsimaritin, all of which possess antioxidant, anti-inflammatory, and anticancer attributes [29]. *E. sandwicensis* is noteworthy for its alkaloids, including erythramine, erysodine, erysopine, and erysovine, as well as novel phytoalexins sandwicarpin and sandwicensin, contributing antimicrobial, anti-inflammatory, anticancer, and antifungal properties [33,34,35,36]. Additionally, *G. brighamii* demonstrates strong antifungal activity against Fusarium species, offering potential as a source of antifungal compounds [39]. *H. arborescens* contains the flavonoid kaempferol, known for its antioxidant, anti-inflammatory, and anticancer properties [41]. However, *H. sandwicensis* presents a more limited flavonoid profile, primarily featuring quercetin 3-*O*-rutinoside, which may be linked to its adaptation to specific ecological niches [42]. *L. yuccoides*, an endemic species, contains alkaloids, including 8-phenylnorlobelol, which exhibit diverse biological activities, including antimicrobial, anticancer, and analgesic effects [43]. The *Lysimachia* genus endemic to Hawaii showcases diverse flavonoid profiles, known for antioxidant, anti-inflammatory, and anticancer properties [45]. *M. barbigera*, an endemic species, contains acetophenones, chromenes, and isomeric compounds, some of which display cytotoxic activity against human ovarian cancer cells [48]. Furthermore, *P. distichus* exhibits potent antimicrobial activity against various bacteria and fungi and holds potential as a therapeutic agent for diabetes mellitus due to its hypoglycemic and antidiabetic properties [51,52,53]. *P. albidus* exhibits antiviral, antibacterial, and potentially anticancer properties, making it a promising subject for scientific investigation [25,54,57,58,61,62]. *P. campanulata* contains alkaloids with documented antiviral and antifungal activity [64]. *P. hawaiiensis* demonstrates antiviral activity against the herpes simplex virus, HIV-1, and various fungal strains, bolstering its value in traditional Hawaiian medicine and as a source of natural anti-infectious agents [25,54]. *R. sandwicensis* contains alkaloids with applications in the pharmaceutical industry [68,69]. *S. paniculatum* leaf extracts exhibit antimicrobial properties, while the essential oil shows promise in repelling red flour beetles and reducing egg hatchability [72,73,74,75]. *S. chrysophylla* presents a diverse range of alkaloids with potential biogenetic pathways [76,77,78]. Hawaiian *Vaccinium* species, especially *V. calycinum* and *V. reticulatum*, are rich in phenolic compounds with antioxidant properties, antimicrobial effects, and potential applications in functional foods and natural preservatives [81,82,83,84]. Moreover, *W. monticola* and *W. uva-ursi* emerge as significant sources of bioactive compounds, with the former containing tumor-inhibitory daphnane diterpenes and the latter showcasing antitumor activity against P-388 lymphocytic leukemia through wikstromol [86,87,88,94]. A study on the leaf of *W. gymnoxiphium* and *W. hobdyi*, exudate flavonoids revealed consistent qualitative profiles with minor quantitative differences [99]. Finally, *Z. dipetalum*, *Z. hawaiiense*, and *Z. kauaense*, three endemic Hawaiian *Zanthoxylum* species, exhibit distinct chemical profiles, including alkaloids, pyranocoumarins, and flavonoids, with potential ecological and pharmaceutical applications [101,102,103,105,106]. The collective findings underscore the importance of preserving Hawaii’s unique biodiversity and exploring the full potential of these native plants for pharmaceutical and ecological purposes.

## 4. Perspectives

The presented findings serve to underscore the diverse and promising pharmacological prospects stemming from bioactive compounds found within endemic Hawaiian plant species. The considerable diversity of these bioactive compounds, including alkaloids, flavonoids, glycosides, and others, found in endemic or native Hawaiian plants, offers a spectrum of bioactive properties encompassing antimicrobial, anti-inflammatory, anticancer, and antioxidant activities. Notably, many of these compounds exhibit considerable potential for incorporation into pharmaceutical applications. For instance, compounds sourced from *A. glauca* var. *glauca*, *Bidens* spp., *C. obovata*, and other species exhibit substantial promise as prospective treatments for a variety of ailments, including cancer, inflammatory disorders, and infectious diseases. Several plant varieties such as *C. persicifolia*, *C. ernodeoides*, and Hawaiian *Vaccinium* species are enriched with flavonoids and allied compounds boasting antioxidant attributes. These compounds hold potential applications in the development of functional foods and natural preservatives. Moreover, the abundance of these bioactive compounds within Hawaii’s endemic plant species underscores the critical need for preserving the state’s unique biodiversity. These plants have thrived in distinct ecological niches and thus present invaluable resources, both from a pharmaceutical and ecological perspective. Certain plants, such as *P. hawaiiensis* and *P. albidus* (Mamaki), have enjoyed historical usage in traditional Hawaiian medicine. The scientific validation of their bioactive properties serves to reaffirm the significance of traditional knowledge and practices. *P. distichus* exhibits potential as a therapeutic agent for diabetes mellitus due to its hypoglycemic and antidiabetic attributes, thereby underscoring its relevance in the management of chronic illnesses. Several plant species, including *C. sandwichianais*, *P. campanulata*, and *P. hawaiiensis*, manifest antiviral and antifungal activities, suggesting their prospective utility as natural anti-infectious agents. The chemical profiles of endemic Hawaiian species, such as *Z. dipetalum*, *Z. hawaiiense*, and *Z. kauaense*, hold promise not solely for pharmaceutical purposes but also for ecological ends, potentially conferring benefits upon Hawaii’s distinct ecosystem. This narrative underscores the significance of sustainable methodologies in harnessing the potential of these native plants, with a dual emphasis on conserving Hawaii’s biodiversity while leveraging their pharmacological promise.

In summation, this review accentuates the abundance of bioactive compounds inherent in Hawaiian endemic plants and their multifaceted potential, encompassing a broad spectrum of pharmacological and ecological applications. Additionally, it underscores the pivotal role these plants may assume in advancing pharmaceutical research, yielding novel medicines, natural remedies, and innovative bioactive compounds, each harboring diverse health benefits. However, it is essential to recognize that further research is imperative to comprehensively elucidate the molecular structures and mechanisms of action of these identified compounds and explore their therapeutic potential. Furthermore, the preservation of these endemic species and their habitats should assume paramount importance to ensure their sustained contributions to human well-being and ecological equilibrium.

## 5. Methodology

The article selection methodology involved systematic searches in prominent scholarly databases, including SciFinder, PubMed, Science Direct, Scopus, Google Scholar, and the Scientific Information Database. English-language articles using specific keywords pertaining to Hawaiian plants, notably endemic Hawaiian plants, phytochemistry, chemical constituents, biological studies, and biological activities, were focused on. Articles investigating the intricate interplay between these plants and their phytochemical components, with due consideration of the impacts of biological research, were prioritized in the selection process. Significantly, subscription-based articles and articles requiring payment for access were deliberately excluded. It is anticipated that the predominant genre of the chosen articles is scholarly reviews.

## Figures and Tables

**Figure 1 ijms-24-16323-f001:**
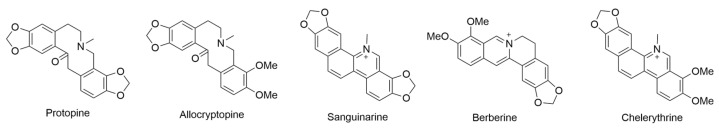
Alkaloids from *A. glauca* var. *glauca*.

**Figure 2 ijms-24-16323-f002:**
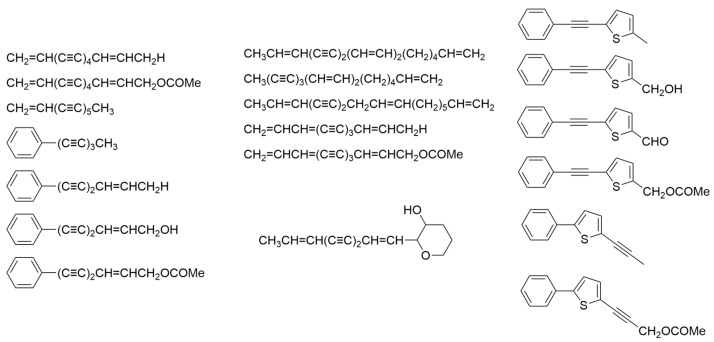
Polyacetylenes from Hawaiian *Bidens* species.

**Figure 3 ijms-24-16323-f003:**
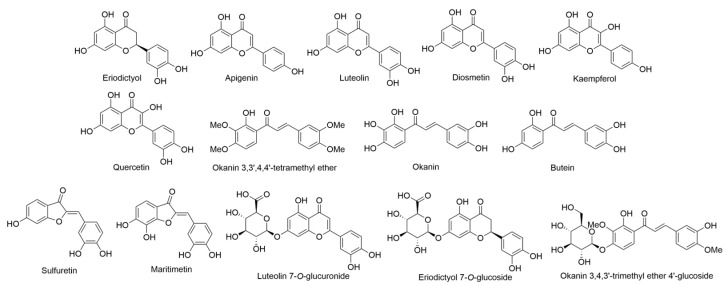
Flavonoids from Hawaiian *Bidens* species.

**Figure 4 ijms-24-16323-f004:**
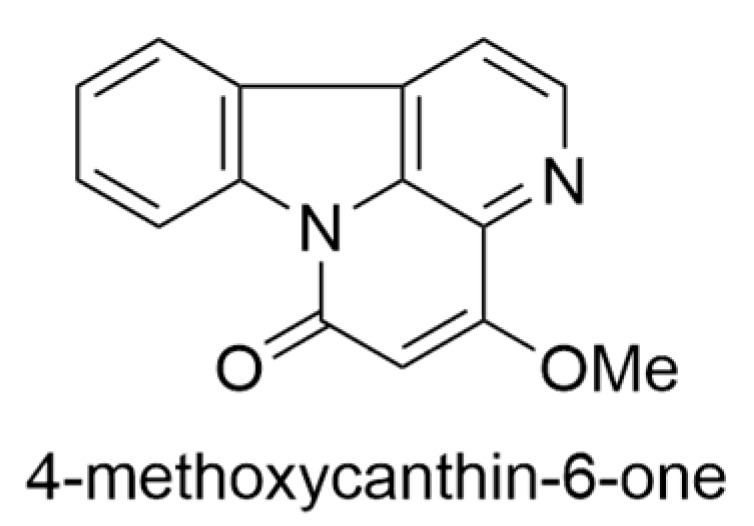
Alkaloid from *C. obovata*.

**Figure 5 ijms-24-16323-f005:**
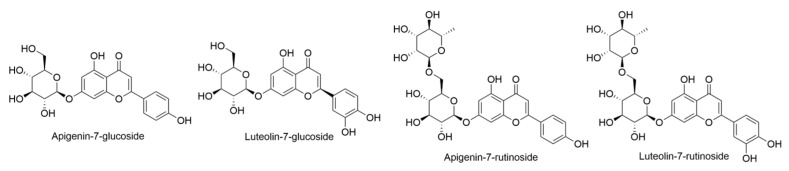
Flavonoids glucoside from *C. persicifolia*.

**Figure 6 ijms-24-16323-f006:**
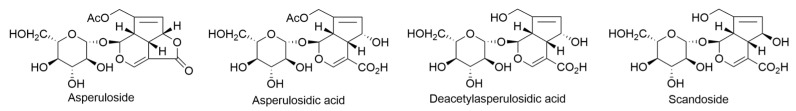
Iridoid glycosides from *C. ernodeoides*.

**Figure 7 ijms-24-16323-f007:**
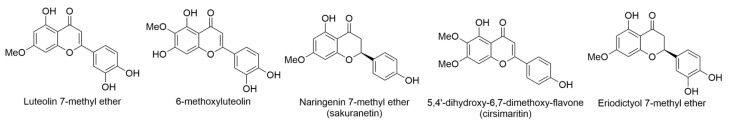
Flavonoids from *D. arborea*.

**Figure 8 ijms-24-16323-f008:**
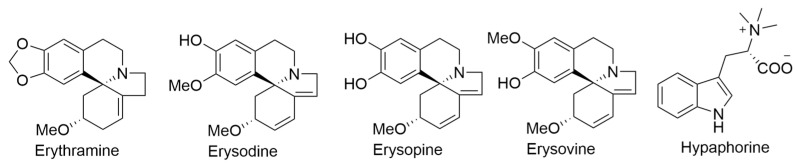
Alkaloids from *E. sandwicensis*.

**Figure 9 ijms-24-16323-f009:**
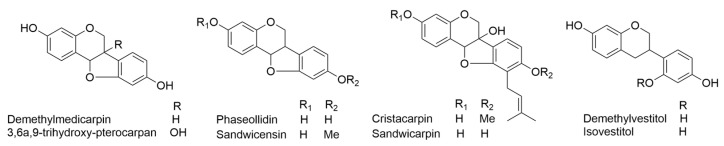
Flavonoids from *E. sandwicensis*.

**Figure 10 ijms-24-16323-f010:**
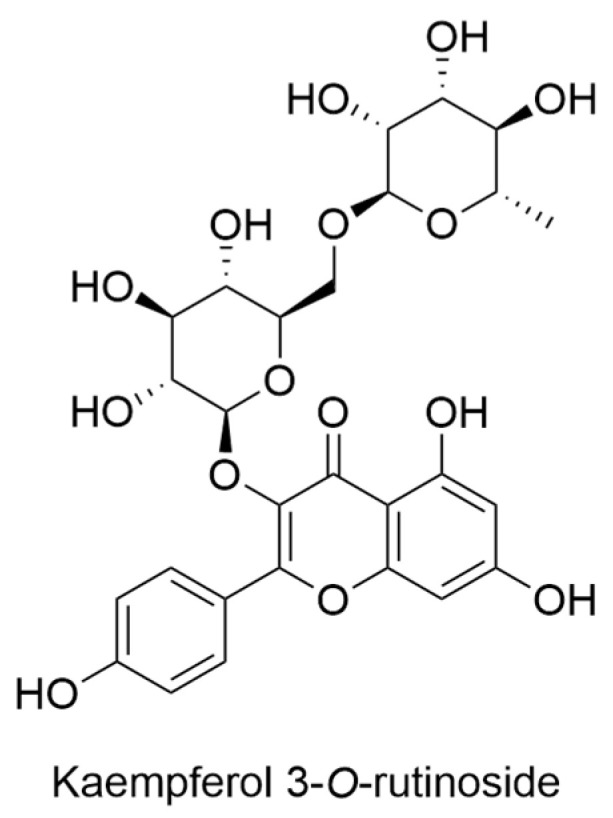
Flavonoid glucoside from *H. arborescens*.

**Figure 11 ijms-24-16323-f011:**
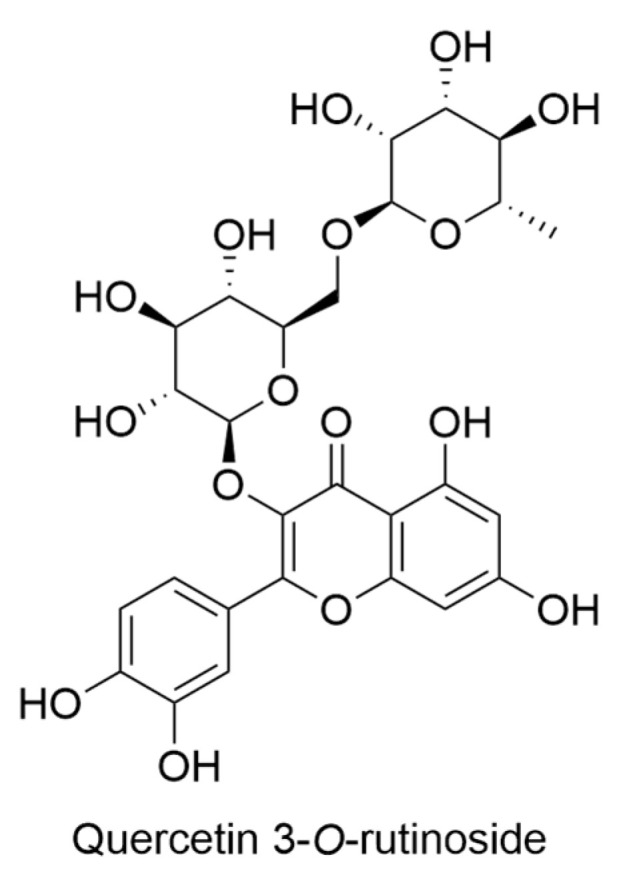
Flavonoid glucoside from *H. sandwicensis*.

**Figure 12 ijms-24-16323-f012:**
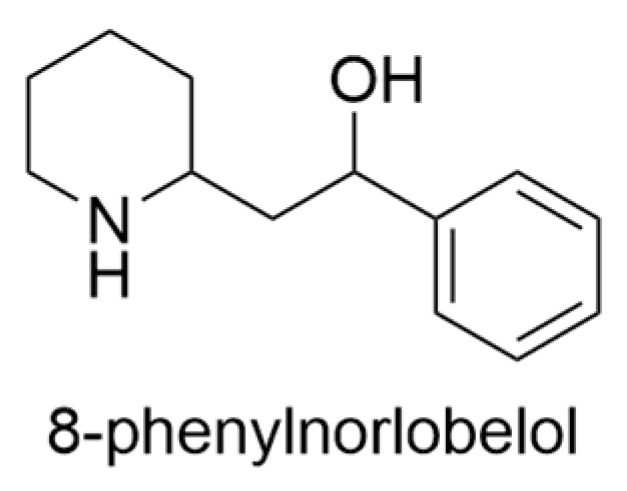
Alkaloid from *L. yuccoides*.

**Figure 13 ijms-24-16323-f013:**
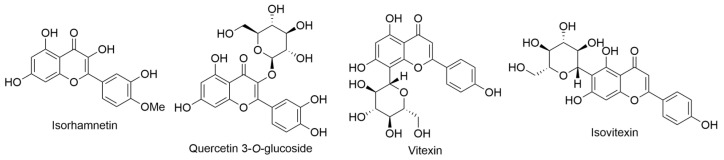
Flavonoids from Hawaiian *Lysimachia* genus.

**Figure 14 ijms-24-16323-f014:**
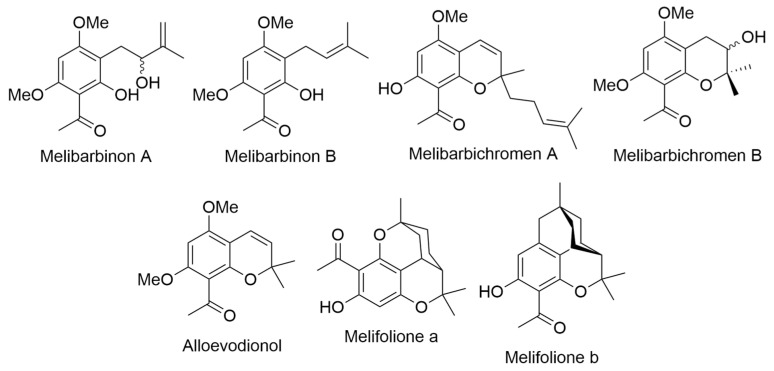
Acetophenones and 2*H*-benzopyranes from *M. barbigera*.

**Figure 15 ijms-24-16323-f015:**
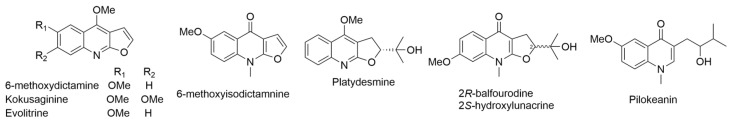
Alkaloids and furoquinolines from *P. campanulatum*.

**Figure 16 ijms-24-16323-f016:**
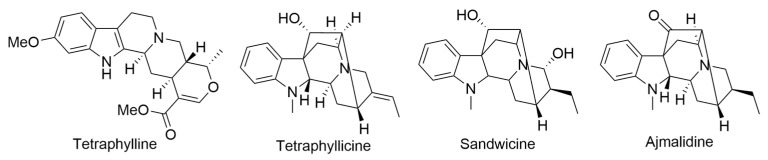
Alkaloids from *R. sandwicensis*.

**Figure 17 ijms-24-16323-f017:**
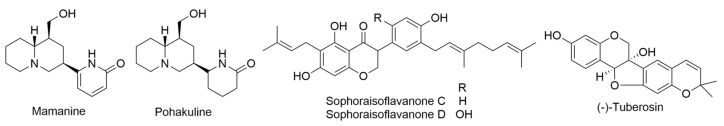
Alkaloids and flavonoids from *S. chrysophylla*.

**Figure 18 ijms-24-16323-f018:**
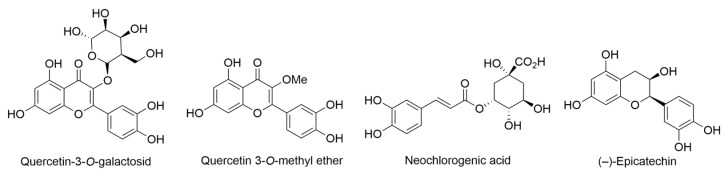
Flavonoid and cinnamate ester from *V. calycinum* and *V. reticulatum*.

**Figure 19 ijms-24-16323-f019:**
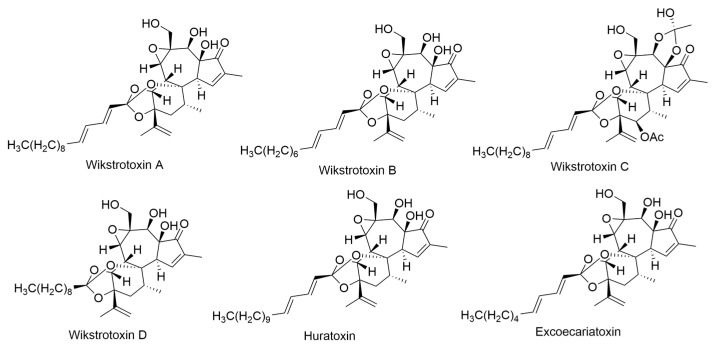
Daphnane diterpenes from *W. monticola*.

**Figure 20 ijms-24-16323-f020:**
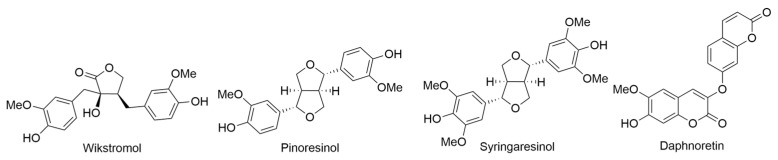
Lignans and coumarin from *W. uva-ursi*.

**Figure 21 ijms-24-16323-f021:**
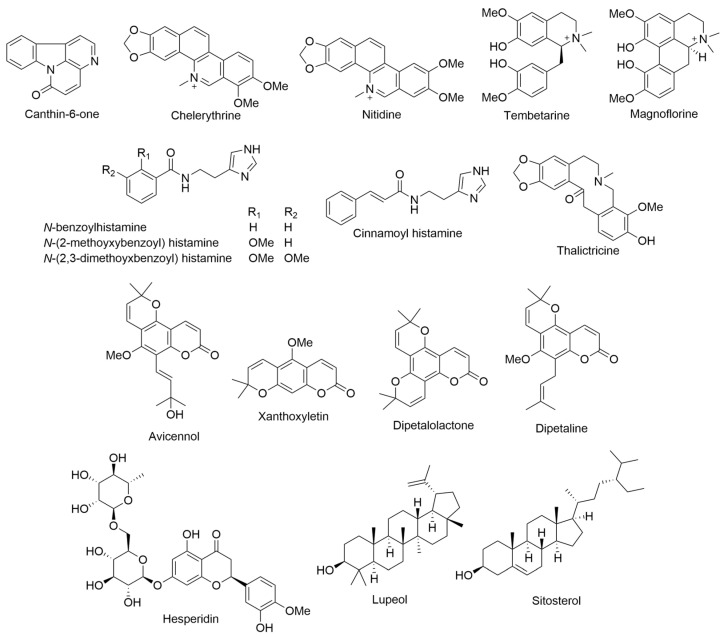
Alkaloids, pyranocoumarins, triterpenes, and flavonoid from *Z. dipetalum*.

## Data Availability

Data sharing is not applicable.

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
