# Peer review of "Phytochemistry and Biological Studies of Endemic Hawaiian Plants"

_ijms, 2023, doi:10.3390/ijms242216323_

Round 1

Reviewer 1 Report

Comments and Suggestions for Authors

Revised text:

This review article provides an extensive overview of the phytochemistry and biological studies of selected endemic Hawaiian plants, emphasizing their medicinal properties and therapeutic potential.  The article effectively incorporates relevant citations, reflecting the authors' diligent work.

Nonetheless, I encourage the authors to incorporate a more critical evaluation of each referenced paper concerning the reported pharmacological activities. For instance, the article mentions various activities associated with the Hawaiian plants under review. To establish the significance of these activities, it's crucial to examine whether appropriate positive controls were employed. Furthermore, a comprehensive evaluation necessitates the inclusion of quantitative data on these activities, permitting comparisons with positive controls. It's advisable that these values employ consistent units for uniformity. The use of subjective descriptors like "potent" or "moderate activity" should be minimized, as these terms often appear in primary literature. Instead, providing quantitative data enables more reliable conclusions through comparisons with proper controls. Additionally, the article should explore the reported toxicity data, specifying the dose levels and how these relate to the activities described in different studies.

When discussing the traditional uses of a plant, it's imperative to clarify which parts of the plant were used and the method of administration. An insightful analysis of the traditional claims should also be undertaken, distinguishing between isolated reports and widely recognized, well-documented traditional practices. This assessment should prioritize the traditional uses that enjoy robust support within the existing literature.

I trust that the authors will embrace this opportunity to improve the review.

Comments on the Quality of English Language

Overall, the article is well-composed, featuring clear and comprehensible language.

Author Response

Dear Reviewer 1,

Thank you for your feedback. We have carefully reviewed your comments and have made the necessary revisions as per your suggestions. Kindly refer to the following responses for your consideration.

Reviewer 1 Comments and Suggestions
Point 1: This review article provides an extensive overview of the phytochemistry and biological studies of selected endemic Hawaiian plants, emphasizing their medicinal properties and therapeutic potential.  The article effectively incorporates relevant citations, reflecting the authors' diligent work.
Response 1: Thank you.

Point 2: Nonetheless, I encourage the authors to incorporate a more critical evaluation of each referenced paper concerning the reported pharmacological activities. For instance, the article mentions various activities associated with the Hawaiian plants under review. To establish the significance of these activities, it's crucial to examine whether appropriate positive controls were employed. Furthermore, a comprehensive evaluation necessitates the inclusion of quantitative data on these activities, permitting comparisons with positive controls. It's advisable that these values employ consistent units for uniformity.
Response 2: Positive controls have been included to determine the significance of the activities, referring to the literature provided. 

Point 3: The use of subjective descriptors like "potent" or "moderate activity" should be minimized, as these terms often appear in primary literature. Instead, providing quantitative data enables more reliable conclusions through comparisons with proper controls. 
Response 3: The available quantitative data has been incorporated. We tried to avoid using subjective descriptors such as "potent" or "moderate activity" too much. These words were “copied” from the original references, most of which also provided quantitative data.

Point 4: Additionally, the article should explore the reported toxicity data, specifying the dose levels and how these relate to the activities described in different studies.
Response 4: The available toxicity data has been examined and included in the article, referred from the literature provided. Please note that not all the plants have been tested for their toxicity, so there were no dose levels and toxicity data available for these plants.

Point 5: When discussing the traditional uses of a plant, it's imperative to clarify which parts of the plant were used and the method of administration. An insightful analysis of the traditional claims should also be undertaken, distinguishing between isolated reports, and widely recognized, well-documented traditional practices. This assessment should prioritize the traditional uses that enjoy robust support within the existing literature.
Response 5: The traditional uses of plants have been derived from the available literature.

Point 6: Overall, the article is well-composed, featuring clear and comprehensible language.
Response 6: Thank you.

Thank you for your consideration.

Sincerely,

Shugeng Cao

Reviewer 2 Report

Comments and Suggestions for Authors

This review by Cao et al. widely describes the main natural compounds isolated from many endemic plant species from the Hawaii Islands, such as alkaloids, polyacetilenes, flavonoids, and others.  The authors indicate in detail the different endemic plants from the zone and the most important natural compounds isolated, following an order based on plant species. The authors also focus on the main biological activities described for the indicated natural products, providing important information about this fact.  

In my opinion, this is an excellent and original work that contributes to the field of natural products studies by providing important information about a wide collection of interesting compounds with possible applications in chemistry, medicine, and pharmacy. This review is well-written and organized, following a well-described methodology. The conclusions indicated by the authors are consistent with the information presented in the review.  

Therefore, I recommend accepting this article after MINOR REVISION:

Some considerations must be taken into account:

- In some parts of the review (for example, page 4), the names of plants are not in cursive letters. The authors have to change the type of letter.

- When the authors described the presence of some compounds in several plants, they do not apport (or only depicted one example, such as in Figures 4, 10, 11, and 12) any schemes with the molecular structures of the mentioned compounds. The authors should include some schemes indicating examples of isolated or mentioned compounds where necessary.

- Some typing mistakes have been detected. A revision is necessary. 

Author Response

Dear Reviewer 2,

Thank you for your feedback. We have carefully reviewed your comments and have made the necessary revisions as per your suggestions. Kindly refer to the following responses for your consideration.

Reviewer 2 Comments and Suggestions
Point 1: In some parts of the review (for example, page 4), the names of plants are not in cursive letters. The authors have to change the type of letter.
Response 1: 
People do not handwrite the plant’s name anymore, so we don’t use cursive letters as others in our manuscript. All names are italicized. The first letter of the genus name is capitalized but the specific epithet is not. Most commonly, italicized font is used when typing the name, and underlining is used when handwriting the name.  Upon the initial mention of the plant's scientific name, the species names will be provided, and subsequent mentions will be abbreviated.

Point 2: When the authors described the presence of some compounds in several plants, they do not apport (or only depicted one example, such as in Figures 4, 10, 11, and 12) any schemes with the molecular structures of the mentioned compounds. The authors should include some schemes indicating examples of isolated or mentioned compounds where necessary.
Response 2: We don’t understand exactly what the reviewer meant. Nevertheless, we have provided structures of the isolated compounds as many as possible. The structures of the mentioned compounds have been separated to enhance readability for the readers. This adjustment is intended to provide a clearer and more organized presentation of the information.

Point 3: Some typing mistakes have been detected. A revision is necessary.
Response 3: All typing mistakes have been checked and revised.

Thank you for your consideration.

Sincerely,

Shugeng Cao

Reviewer 3 Report

Comments and Suggestions for Authors

In this manuscript entitled “Phytochemistry and biological studies of the endemic Hawaiian plants”, the authors provide an in-depth analysis of the phytochemistry and biological studies of selected endemic Hawaiian plants, highlighting their medicinal properties and therapeutic potential. This review emphasizes the rich phytochemical diversity and biological activities found in endemic Hawaiian plants, showcasing their potential as sources of novel therapeutic agents, which is important for scientific exploration, conservation, and sustainable utilization of these valuable resources. In general, this manuscript can be accepted for publication in this journal. However, there are still several issues to be solved be addressed.

1. In Figure 12. The compound of 8-phenylnorlobelol contains two chiral centers, the author did not write, please explain.

2. It is recommended that the author summarize the parts containing flavonoids, such as the sections 2.4, 2.8, 2.11, 2.12, and 2.14.

3. As a review, the comprehensiveness of the content is very important, and it is suggested that the author should add common or representative extraction and separation methods in order to better comprehensive utilization of plants with high added value.

Author Response

Dear Reviewer 3,

Thank you for your feedback. We have carefully reviewed your comments and have made the necessary revisions as per your suggestions. Kindly refer to the following responses for your consideration.

Reviewer 3 Comments and Suggestions
Point 1: In Figure 12. The compound of 8-phenylnorlobelol contains two chiral centers, the author did not write, please explain.
Response 1: The literature does not provide information on the stereochemistry of the compound 8-phenylnorlobelol. As a result, the configuration of the two chiral centers remains unspecified in the available sources.

Point 2: It is recommended that the author summarize the parts containing flavonoids, such as the sections 2.4, 2.8, 2.11, 2.12, and 2.14.
Response 2: The sections containing information on flavonoids, including sections 2.4, 2.8, 2.11, 2.12, and 2.14, have been provided in the discussion section. Additionally, a summarized overview can be found in Table S2, which is referenced from the existing literature provided.

Point 3: As a review, the comprehensiveness of the content is very important, and it is suggested that the author should add common or representative extraction and separation methods in order to better comprehensive utilization of plants with high added value.
Response 3: Extraction and separation methods, referenced from existing literature, have been added, referred from the literature provided.

Thank you for your consideration.

Sincerely,

Shugeng Cao